# Four new species of *Pristimantis* Jiménez de la Espada, 1870 (Anura: Craugastoridae) in the eastern Amazon

**Elciomar Araújo de Oliveira**[1,¤]*, **Leandro Alves da Silva**[2], **Elvis Almeida Pereira Silva**[3,4,5], **Karen Larissa Auzier Guimarães**[6,7], **Marcos Penhacek**[8], **José Gregório Martínez**[9], **Luís Reginaldo Ribeiro Rodrigues**[6,7], **Diego José Santana**[4], **Emil José Hernández-Ruz**[10]

**1** Programa de Pós-Graduação em Biodiversidade e Biotecnologia da Rede BIONORTE, Universidade Federal do Amazonas, Manaus, Brazil, **2** Departamento de Sistemática e Ecologia, Centro de Ciências Exatas e da Natureza, Universidade Federal da Paraíba, João Pessoa, Paraíba, Brazil, **3** Laboratório de Herpetologia, Departamento de Biologia Animal, Programa de Pós-Graduação em Biologia Animal, Universidade Federal Rural do Rio de Janeiro, Seropédica, Rio de Janeiro, Brazil, **4** Mapinguari—Laboratório de Biogeografia e Sistemática de Anfíbios e Répteis, Universidade Federal de Mato Grosso do Sul, Cidade Universitária, Mato Grosso do Sul, Brazil, **5** Technische Universität Braunschweig, Zoological Institute, Braunschweig, Alemanha, **6** Programa de Pós-graduação em Recursos Naturais da Amazônia, Universidade Federal do Oeste do Pará, Santarém, Pará, Brazil, **7** Laboratório de Genética & Biodiversidade, Instituto de Ciências da Educação, Universidade Federal do Oeste do Pará, Santarém, Pará, Brazil, **8** Instituto de Ciências Naturais, Humanas e Sociais e Acervo Biológico da Amazônia Meridional– ABAM, Universidade Federal de Mato Grosso, Sinop, Mato Grosso, Brazil, **9** Grupo de Investigación Biociencias, Facultad de Ciencias de la Salud, Institución Universitaria Colegio Mayor de Antioquia, Medellín, Colombia, **10** Laboratório de Zoologia Adriano Giorgi, Programa de Pós-Graduação em Biodiversidade e Conservação, Campus Universitário de Altamira, Universidade Federal do Pará, Altamira, Pará, Brazil

☯ These authors contributed equally to this work.
¤ Current address: Laboratório de Zoologia Adriano Giorgi, Faculdade de Ciências Biológicas, Universidade Federal do Pará, Altamira, Pará, Brazil
* elciomar.atractus@gmail.com

**Data Availability Statement:** All relevant data are within the manuscript and its Supporting Information files.

## Abstract

The *Pristimantis* genus (Anura: Craugastoridae) is the most diverse among all vertebrates with 531 described species. The highest diversity occurs in Ecuador (215 species), followed by Colombia (202), Peru (139), Venezuela (60), Brazil (30), Bolivia (17), Guyana (6) Suriname and French Guiana (5). The genus is divided into 11 species groups. Of these, the *P. conspicillatus* group (containing 34 species), distributed in extreme southeastern Costa Rica, Isla Taboga (Panama), northern South America (from Colombia to eastern Guyana), south Bolivia, and is the best represented in Brazil (16 species). The main characteristics of this group are the tympanic membrane and tympanic annulus distinct (except in *P. johannesdei*); dorsum smooth or shagreen; dorsal lateral fold present or absent; usually smooth belly, but may be weakly granular in some species; toe V slightly larger than the toe III. Most of the taxonomic inconsistencies in species of *Pristimantis* could be due to its much conserved morphology and the lack of comprehensive taxonomic evaluations. Thus, an ongoing challenge for taxonomists dealing with the *Pristimantis* genus is the ubiquitous abundance of cryptic species. In this context, accurate species delimitation should integrate evidences of morphological, molecular, bioacoustics and ecological data, among others. Based on an integrative taxonomy perspective, we utilize morphological, molecular

**Funding:** Thanks to Capes, according Edital 143/ 2019, item 11.1, b) "This study was financed in part by the Coordenação de Aperfeiçoamento de Pessoal de Nível Superior - Brasil (CAPES) - Finance Code 001". EAO was supported by the "Programa de Apoio à Produção Qualificada" - PAPQ/UFPA (Proc.N. 23073.026959/2016-92) and received PhD scholarship from Conselho Nacional de Desenvolvimento Científico and Tecnológico (CNPq/Brazil-www.cnpq.br http://brazil-www.cnpq. br) (Proc.N. 141718/2016-1). LAS thanks to CNPq by the scholarship (140408/2018-5). KLAG received Mastership from Coordenação de Aperfeíîoamento de Pessoal de Nível Superior (CAPES/Brazil-www.capes.gov.br http://brazil-www.capes.gov.br/>). DJS thanks CNPq for his research fellowship (311492/2017-7). LRRR received grants from CAPES/ProAmazônia (AUXPE 3318/2013). The funders had no role in study design, data collection and analysis, decision to publish, or preparation of the manuscript.

**Competing interests:** The authors have declared that no competing interests exist.

(mtDNA) and bioacoustic evidence to describe four new species of the *Pristimantis conspicillatus* group from the eastern Amazon basin. *Pristimantis giorgii* **sp. nov.** is known from the Xingu/Tocantins interfluve and can be distinguished from the other *Pristimantis* species of the region by presenting discoidal fold, dorsolateral fold absent, vocalization composed of three to four notes and genetic distance of 7.7% (16S) and 14.8% (COI) from *P. latro*, the sister and sympatric species with respect *P. giorgii* **sp. nov.**. *Pristimantis pictus* **sp. nov.** is known to the northern Mato Grosso state, Brazil, and can be distinguished from the other species of *Pristimantis* by presenting the posterior surface of the thigh with light yellow patches on a brown background, also extending to the inguinal region, vocalization consisting of four to five notes and a genetic distance of 11.6% (16S) and 19.7% (COI) from *P. pluvian* **sp. nov.**, which occurs in sympatry. *Pristimantis pluvian* **sp. nov.** is known to the northern Mato Grosso state, Brazil, and may be distinguished from the other *Pristimantis* species by having a posterior surface of the thigh reddish and vocalization composed of two notes. *Pristimantis moa* **sp. nov.** is known to the northern Tocantins state and southwestern Maranhão state. This species can be distinguished from the other *Pristimantis* species by possessing slightly perceptible canthal stripe, external thigh surface with dark yellow spots on brown background, vocalization consisting of three to five notes and genetic distance of 2.3–11.7 (16S) and 10.5–23.1 (COI) for the new *Pristimantis* species of this study.

## Introduction

The genus *Pristimantis* Jiménez de la Espada 1870, is the most diverse among vertebrates [1], currently comprised of 531 valid species, 155 of which have only been described in the last decade [2]. Most species of this genus occur in the western part of the Amazon [3]. The topographic heterogeneity of the Andes, accompanied by the life history of *Pristimantis* (direct development, high endemism and the ability to colonize a wide variety of habitat types, including highlands) could be partially responsible for this great diversity [4]. In the eastern Amazon, few studies have focused on taxonomic aspects of *Pristimantis* populations, and most of the information about this genus comes from faunal inventories [5–8], of which many do not present specific identification of existing lineages [9].

The genus is currently divided into 11 morphologically defined groups of species [10]. The *P. conspicillatus* group contains 34 species distributed mainly in northern South America (from Costa Rica to eastern Guyana and Taboga Islands), south Bolivia and Atlantic forest, Brazil [11]. The main characteristics of this group are the tympanic membrane and tympanic annulus distinct (except in *P. johannesdei* Rivero & Serna 1988); dorsum smooth or shagreen; lateral dorsal fold present or absent; usually smooth belly, but may be weakly granular in some species; toe V slightly larger than the toe III (for more details see Hedges and colleagues [11]. In Brazil, 16 species of the aforementioned group are known: *Pristimantis buccinator* (Rodriguez, 1994), *P. chiastonotus* (Lynch & Hoogmoed, 1977), *P. conspicillatus* (Gunther, 1858), *P. dundeei* (Heyer & Muñoz, 1999), *P. fenestratus* (Steindachner, 1864), *P. gutturalis* (Hoogmoed, Lynch & Lescure, 1977), *P. latro* de Oliveira, Rodrigues, Kaefer, Pinto, and Hernández-Ruz 2017, *P. malkini* (Lynch, 1980), *P. paulodutrai* (Bokermann, 1975), *P. peruvianus* (Melin, 1941), *P. ramagii* (Boulenger, 1888), *P. skydmainos* (Flores & Rodriguez, 1997), *P. ventrigranulosus* (Maciel, Vaz-Silva, Oliveira & Padial, 2012), *P. vilarsi* (Melin, 1941), *P. vinhai* (Bokermann, 1975) and *P. zeuctotylus* (Lynch & Hoogmoed, 1977).

The species of the *P. conspicillatus* group mentioned above, only *P. gutturalis*, *P. chiastonotus*, *P. zeuctotylus*, *P. latro* and *P. dundeei* occur in the eastern Amazon rainforest. Adding *P.*

*inguinalis*, *P. marmoratus* (Boulenger, 1900), and *P. zimmermanae* (Heyer & Hardy, 1991) are recorded eight species of *Pristimantis* to the east of the Brazilian Amazon. This region has been suffering from strong deforestation pressures in recent years, causing habitat loss, which is one of the main threats to biodiversity [12], [13].

Most of the taxonomic inconsistencies within *Pristimantis* can be attributed to its conserved morphology and its high diversity [1], [9], [14]. The conserved morphology of the species is seen in the high cryptic diversity within the genus, where distinct lineages are usually associated with the same nominal species [15], [16]. Given this context, correctly delimiting a species often requires an integrative approach, associating morphological, molecular, and bioacoustic data [14], [17], [18]. After several expeditions in different regions of the eastern Amazon and north Cerrado, we recorded *Pristimantis* lineages that we could not associate with any nominal species. Based on an integrative taxonomy framework, herein we describe four new *Pristimantis* species belonging to the *P. conspicillatus* group.

## Materials and methods

### Ethics statement

The permits for field samples collections were provided by the Brazilian Government (SISBIO licence N 30034–1, 32401, 54493–5, and 54493–11). Although euthanized methods were not evaluated by an institutional animal care and use committee or similar regional ethics committee, voucher collections strictly complied with the ethical conditions as dictated by the Sistema of Autorização and Informação in Biodiversidade–SISBIO, environmental authorities of Brazil (see scientific license numbers above). The collected frogs were euthanized by topical application of a 2% liquid solution of lidocaine hydrochloride, fixed in 10% formalin and transferred to permanent collections in alcohol 70%. We collected tissue samples (muscle and liver) before individuals were fixated and stored.

### Nomenclatural acts

The electronic edition of this article conforms to the requirements of the amended International Code of Zoological Nomenclature, and hence the new names contained herein are available under that Code from the electronic edition of this article. This published work and the nomenclatural acts it contains have been registered in ZooBank, the online registration system for the ICZN. The ZooBank LSIDs (Life Science Identifiers) can be resolved and the associated information viewed through any standard web browser by appending the LSID to the prefix "http://zoobank.org/". The LSID for this publication is: urn:lsid:zoobank.org:pub:A1A7A538-68C8-401B-B706-6C7C16D87296. The electronic edition of this work was published in a journal with an ISSN, and has been archived and is available from the following digital repositories: PubMed Central, LOCKSS.

### Sampling

We conducted active searches in 27 localities, including areas typical to the Amazonian domain (Pará state and northern Mato Grosso state), transition areas between the Amazon and Cerrado (Araguaína, in Tocantins state and Balsas, in Maranhão state) and Cerrado areas (Riachão in Maranhão state and Palmas in Tocantins state) (Fig 1). Additionally, we analyzed 291 specimens from the following zoological collections: Coleção Zoológica de Referência da Universidade Federal do Mato Grosso do Sul (ZUFMS-AMP), Acervo Biológico da Amazônia Meridional (ABAM) da Universidade Federal do Mato Grosso (UFMT/Sinop), Coleção de Anfíbios e Répteis do Instituto Nacional de Pesquisas da Amazônia (INPA-H), Coleção do

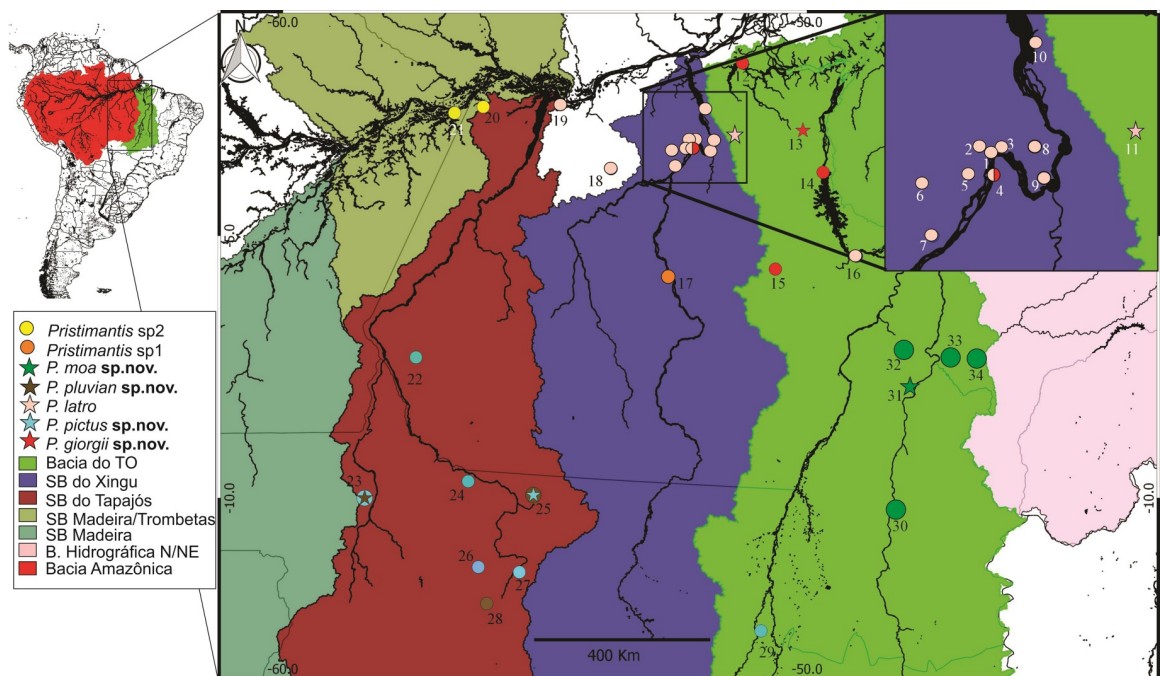

**Fig 1. Map of the collection points and distribution of the new *Pristimantis* species in eastern Amazon.** Abbreviations: SB = Sub Basins; PA = State of Pará; AM = State of Amazonas; TO = State of Tocantins; MA = Maranhão State; MT = Mato Grosso state. 1–5) Altamira—PA; 6–7) Brasil Novo, PA; 8–9) Volta Grande, PA; 10) Senador José Porfírio, PA; 11) Anapu, PA; 12) Caxiuanã, PA; 13) Portel, PA; 14) Tucuruí, PA; 15) Carajás, PA; 16) Marabá, PA; 17) Terra do Meio, PA; 18) Medicilândia, PA; 19) Santarém, PA; 20) Juruti, PA; 21) Parintins, AM; 22) Jacareacanga, PA; 23) Cotriguaçu, MT; 24) Paranaíta, MT; 25) Novo Mundo, MT; 26) Tabaporã, MT; 27) Sinop, MT; 28) Novo Mundo, MT; 29) Cristalino, MT; 30) Palmas; 31) Palmeirante, TO; 32) Araguaína, TO; 33) Carolina, MA and 34) Riachão, MA. Author: Elciomar A. de Oliveira).

Museu Paraense Emílio Goeldi (MPEG), Coleção Zoológica da Universidade Federal do Oeste do Pará (UFOPA), Coleção Zoológica da Universidade Federal de Goiás (ZUFG) e Coleção do Laboratório de Zoologia Adriano Giorgi (LZAG) na Universidade Federal do Pará/Altamira (UFPA/Altamira).

## Molecular analysis

We extracted the total genomic DNA using the 2% CTAB protocol [19]. We amplified a fragment of 16S mtDNA by polymerase chain reaction (PCR) using *16Saf* primers and *16Sbr* [20], and the *Cytochrome Oxidase sub unit I* (*COI*) with the primers dgLCO 1490 *forward* and dgHCO 2198 *reverse* [21]. The sequences were obtained through the di-desoxiterminal Sanger method using ABI PRISM® Big Dye™ Terminator V.3 Cycle Sequencing kit (Applied Biosystems™), with 16Saf and COI (forward and reverse). PCR details and sequencing reaction are available in Supporting information (S1 File).

GenBank sequences (https://www.ncbi.nlm.nih.gov/genbank/) of other *Pristimantis* species and of the *P. conspicillatus* group were used in the database for phylogenetic analyses (access numbers are listed in S2 File). The sequences were aligned using the ClustalW Algorithm [22] implemented in the software Geneious v. 9.1.2. The alignment of the 16S mtDNA gene presented gaps, which were removed using GBLOCKS v.0.91b [23], [24], available online (http://molevol.cmima.csic.es/castresana/Gblocks_server.html). A gene tree was built using the MrBayes software [25], with the concatenated genes (16s + COI) using the evolutionary model HKY + G, estimated in the JModelTest software [26] under the Bayesian information criterion

(BIC). We followed the standard values for two runs and four chains, running $50^7$ generations, with a tree sampled every 10.000 generations. The burn in value was selected by viewing the log likelihoods associated with the distribution of the tree after the software Tracer v 1.5 [27], discarding 20% of the trees with the TreeAnnotator software v1.8 [28]. We accessed convergence by examining the average standard deviation of the split frequency between runs ($< 0.01$). The number of independent samples was considered sufficient when the stationary was reached and the Effective Sample Sizes (ESS) were higher than 200. As additional evidence for species delimitation, the uncorrected pairwise distance (p-distances for the 16S rRNA and Kimura 2 Parameters–K2P for the COI) among the species of *Pristimantis* of this study and others in the *P. conspicillatus* group were calculated using MEGA 6.0 [29]. Hedges and colleagues [11] suggest *Oreobates* genus as sister group of *Pristimantis*, so we used it as the outgroup (*O. quixensis*s of Amazonas State) in our analysis. We built a network of haplotypes among *Pristimantis* species from the 16S mtRNA and COI mtDNA genes in the POPART software [30] using the median-joining network method in order to check for possible haplotype sharing.

## Morphological analysis

The nomenclature of morphological characters has been described following the current literature [14], [31–33]: 1) belly skin texture (smooth, granular, granular posterolaterally); 2) dorsal tubercle (presence/absence of dorsal tubercles or short folds); 3) fringe on the fingers (prominent, weak, absent); 4) dorsalateral fold (prominent or weak); 5) fringe on toes (prominent, weak or absent); 6) basal webbing between toes (present or absent); 7) Inner tarsal fold (long, short or absent); 8) color pattern of the throat, chest and belly (heavily stained, weakly stained, immaculate); 9) pattern of the posterior surface of the thigh.

Measurements were made using a 0.01 mm digital caliper and rounded to 0.1 mm as in [31–34]. The measures obtained were: snout-vent length (from tip of snout to posterior margin of vent), SVL; head length (from posterior margin of lower jaw to tip of snout), HL; head width, (at level of angle of jaw ), HW; snout length (from anterior corner of eye to tip of snout), SL; distance from eye to nostril (from anterior corner of eye to posterior margin of naris), DEN; internarial distance (taken between the median margins of the nares), ID; eye length (measured horizontally), EL; interorbital distance (taken between the inner margins of the orbits), IoD; eyelid width (the largest transverse width of the upper eyelid.), EW; tympanum length (the largest length of the tympanum from the anterior margin to the posterior margin of the tympanum), TL; arm length (from the tip of the elbow to the proximal edge of the palmar tubercle), AL; hand length (from the proximal edge of the palmar tubercle to the tip of finger III), HaL; thigh length (from vent to knee), ThL; tibia length (from outer edge of flexed knee to heel), TiL; tarsus length (from the heel to the proximal edge of the inner metatarsal tubercle.), TaL; foot length (from proximal border of inner metatarsal tubercle to tip of fourth toe), FL; leg length (from the knee joint to the tip of toe IV), LL (all morphometric measurements are listed in S3 File). Sex was determined by direct visualization of secondary sexual characters in adult individuals, such as the presence or absence of vocal slits, vocal sac and nuptial callus in males. Comparisons of characters states were performed with specimens from the *P. conspicillatus* group available in specialized literature [31], [32], [34–37].

## Bioacoustic analysis

We used two types of audio recorders to capture vocalizations: Sony Digital Recorder ICD-PX240 and Tascam DR-40 digital recorder. We obtained records of advertisement calls (vocalizations) from seven males between 18h – 20h, at Uberlândia Farm, Portel municipality, Pará state; four males from Jacareacanga municipality, Pará state; two males from Paranaíta

municipality, Mato Grosso state; three males from Taquaruçu district, Palmas municipality, Tocantins state; and five males from Balsas municipality, Maranhão state. The species of the *P. conspicillatus* group selected for acoustic comparisons were *P. fenestratus*, *P. zeuctotylus*, *P. chiastonotus*, *P. samaipatae* [38], *P. vilarsi*, *P. gutturalis* and *P. latro*, as they present similar morphological and geographic distribution (east of the Amazon) as the species described in this study, except for *P. vilarsi* (western Amazon) and *P. samaipatae* (Bolivia).

We analyzed vocalizations at a sampling rate of 44.1 kHz and 16-bit resolution using the Raven Pro v 1.5 software for Windows (Bioacoustics Research Program 2014). Information about frequency was obtained through Fast Fourier Transformations (FFT; width of 1024 points), Frame = 100, Overlap = 75 and flat top filter. Spectrograms and oscillograms were generated using the SeeWave package [39] in the R software (R Core Team, 2018) following the same parameters as the Raven software. The following variables were measured according to specialized literature [34], [40]: call length (ms), number of notes per call, length of the note (ms), presence of pulses, fundamental frequency (frequency band in which the first sound is visualized through a spectral slice output, in Hz) and dominant frequency (measured from a spectral slice taken from the highest amplitude portion of the note, in Hz). The vocalizations were deposited in the Fonoteca Mapinguari UFMS (MAP-V 306–312 *Pristimantis giorgii* **sp. nov.**; MAP-V 313–316 *P. pictus* **sp. nov.**; MAP-V 317–319 *P. pluvian* **sp. nov.**; and MAP-V 320–325 *P. moa* **sp. nov.**).

## Species delimitation

To delimit the Candidate Species (CS), we followed the model proposed by Padial and colleagues (2010) [41]: where a given CS can be classified as an Unconfirmed Candidate Species (UCS), Confirmed Candidate Species (CCS) and Deeply Conspecific Lineages (DCL). UCS is considered any CS that presents genetic differences above the limit ($>$ 3%) proposed for the 16S rRNA gene [18], [42], [43], but with no complementary characters verified, such as morphology, bioacoustics, ecology or distribution. A CCS is considered a CS that presents genetic differentiation in relation to other species and also has support of parallel evidence. Finally, a DCL is considered a CS that presents genetic divergence above the proposed limit ($>$ 3%), but cannot be differentiated by parallel evidence (morphology and acoustics).

## Results

### Phylogenetic relationships and genetic distances

Our tree topology, based only on 16S and COI genes (82 sequences of two concatenated mitochondrial genes, 866bp), recovered posterior probabilities support values among three main divergent lineages.(Fig 2): 1) Andean, 2) Amazon Basin, and 3) Tocantins Basin. The Andean lineages were represented by *P. peruvianus*, *P. achatinus*, *P. vilarsi* and *P. bipunctatus* [44]. The Amazon Basin lineages consists in *Pristimantis zeuctotylus*, *P. fenestratus* complex (with three cryptic lineages, one for Manaus municipality and two for Borba municipality in Amazonas state), *Pristimantis* sp1 (Altamira municipality), *Pristimantis* sp2 (Juruti municipality), *P. latro*, and two undescribed lineages for Mato Grosso state. Two lineages were found for the Tocantins Basin, one of which occurs in the eastern Pará state and another in Tocantins and Maranhão states.

The two undescribed *Pristimantis* lineages from the Amazon Basin occur in northern Mato Grosso (Cristalino, Novo Mundo, Paranaíta, Sinop, Tabaporã and Cotriguaçu municipalities), showing genetic distances (Table 1) between each other of 11.6% (gene 16S rRNA) and 25.9% (gene COI mtDNA) and distant phylogenetic positioning, even though being sympatric in some localities (e.g. Cotriguaçu and Novo Mundo). The lineage identified as *Pristimantis* sp1,

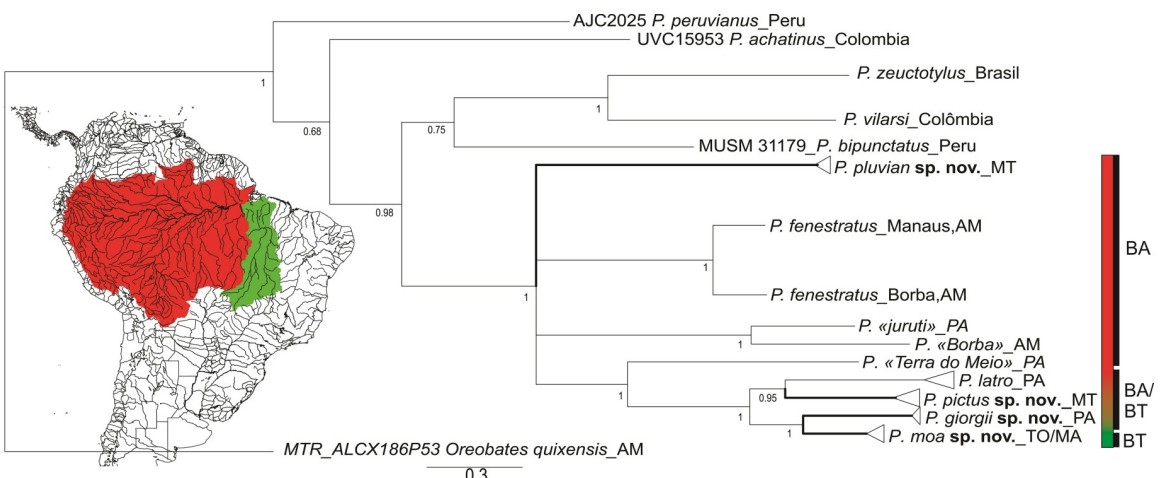

**Fig 2. A gene tree reconstructed by Bayesian analysis using the evolutionary model HKY + G with the 16S and COI genes concatenated (866 bp) between the new species of *Pristimantis* in the eastern Amazonia and others from the *P. conspicillatus* group closest genetically and geographically.** The values below the nodes correspond to the posterior probability. The bar below represents the number of substitutions per site. The red color in the left hand corner represents the Amazon Basin (BA in the right corner of the tree) and the green color represents the Tocantins Basin (BT in the right corner of the tree).

with individuals from the left bank of the upper Xingu River, in Altamira municipality, represents the sister group of the clade formed by *P. latro* and three undescribed *Pristimantis* lineages from the Amazon Basin and Tocantins (HPD = 1), with genetic distances of 9.1–9.9% (16s rRNA gene) and 19.5–22% (COI mtDNA gene). *Pristimantis* sp2, represented by individuals from the municipality of Juruti, Pará state, were found as a sister group of *P. fenestratus* (lineage from Borba, Amazonas state, support HPD = 1), with genetic distances of 4% (16S rRNA gene) and 14.6% (COI mtDNA gene).

In the Tocantins Basin, the two lineages found showed genetic distances of 2.3% (16S rRNA gene) and 10.5% (COI mtDNA gene) among themselves, with one found only in Pará state and the other in Tocantins and Maranhão states. Regarding phylogenetic positioning, the two lineages are sisters with high support (HPD = 1). *Pristimantis latro* (occurring in the Amazon Basin and Tocantins Basin), is sympatric with the Pará lineage, presenting genetic distances of

**Table 1. Uncorrected distance genetics (%) between the new species of *Pristimantis* and other species of the *P. conspicillatus* group of this study.** In below diagonal distance *p* (16S) and above diagonal Kimura 2 Parameters (COI). 1) *Pristimantis moa* **sp. nov.** (Tocantins and Maranhão); 2) *P. giorgii* **sp. nov.** (Pará state); 3) *P. pictus* **sp. nov.** (Mato Grosso and Pará state); 4) *P. latro* (Pará state); 5) *P. fenestratus* (Amazonas state); 6) *P. fenestratus* (Amazonas state); 7) *Pristimantis* sp1 (Pará state); 8) *Pristimantis* sp2 (Pará state); 9) *P. pluvian* **sp. nov.** (Mato Grosso state); 10) *P. fenestratus* (Amazonas state); 11) *P. bipunctatus* (Peru).

| | 1 | 2 | 3 | 4 | 5 | 6 | 7 | 8 | 9 | 10 | 11 |
|---|---|---|---|---|---|---|---|---|---|---|---|
| 1 | - | 10.5 | 12.5 | 12.2 | 24.6 | 24.4 | 19.7 | 24.8 | 21.2 | 23.1 | 25.6 |
| 2 | 2.3 | - | 13.6 | 14.8 | 23.8 | 24.1 | 20.7 | 25.8 | 21.0 | 27.7 | 27.1 |
| 3 | 4.7 | 6.2 | - | 12.4 | 23.8 | 24.5 | 19.5 | 24.6 | 19.7 | 25.9 | 24.4 |
| 4 | 6.2 | 7.7 | 4.6 | - | 23.9 | 24.0 | 22.0 | 22.7 | 20.6 | 24.4 | 24.8 |
| 5 | 8.2 | 8.8 | 7.6 | 8.7 | - | 6.2 | 23.3 | 23.9 | 20.2 | 25.0 | 25.2 |
| 6 | 8.4 | 9.2 | 8.0 | 8.1 | 1.9 | - | 21.7 | 22.4 | 21.1 | 24.8 | 25.4 |
| 7 | 9.5 | 9.7 | 9.9 | 9.1 | 11.2 | 11.3 | - | 20.6 | 21.9 | 23.6 | 24.0 |
| 8 | 11.6 | 12.0 | 11.2 | 12.4 | 12.2 | 12.8 | 14.5 | - | 22.2 | 14.6 | 24.0 |
| 9 | 11.7 | 12.5 | 11.6 | 10.7 | 11.6 | 11.7 | 13.2 | 13.4 | - | 21.6 | 19.5 |
| 10 | 12.7 | 13.3 | 12.1 | 13.2 | 12.3 | 12.9 | 15.3 | 4.0 | 14.4 | - | 23.8 |
| 11 | 13.6 | 14.3 | 13.1 | 11.9 | 11.7 | 11.4 | 13.7 | 13.8 | 15.4 | 14.6 | - |

7.7% (16S rRNA gene) and 14.8% (COI mtDNA gene) and is sister to the lineages from northern Mato Grosso state (HPD = 0.95).

The haplotype networks for the 16S (422 bp) and COI (444 bp) genes showed seven distinct mitochondrial haplogroups, corroborating the lineages found in phylogenetic reconstruction, with no haplotype sharing among the new species of *Pristimantis* described in this study (Fig 3). Based on the large genetic distance found with the 16S and COI mitochondrial markers, phylogenetic positioning, and morphological and acoustic (Fig 4) divergences (mentioned in the "*Diagnosis*" and *"Comparison with other species*" sections), herein we describe four new species occurring throughout the east of Amazon and north Cerrado.

## Systematics

**Allocation to species group.** The new taxon described in this article belong to the *Pristimantis conspicillatus* group based on (1) molecular phylogenetic relationship (Fig 2); and (2) morphological characteristics such as: tympanic membrane and tympanic annulus distinct (except in *P. johannesdei*); dorsum smooth or shagreen; lateral dorsal fold present or absent; usually smooth belly, but may be weakly granular in some species; toe V slightly larger than toe III (see Hedges and colleagues [11]).

*Pristimantis giorgii* **sp. nov.** (Fig 5A–5D) urn:lsid:zoobank.org:act:7CA1240B-0506-45F9-A9E1-0E38A18A0B65

**Holotype.** (Fig 6A–6D), LZAG 1381, adult male collected on 06 February 2018 in Portel municipality, Pará state, Brazil (02˚59'38.10" S; 50˚4'48.57" W) by Elciomar Araújo de Oliveira, Emil José Hernández Ruz and Erick Fabrício Santos Moraes. Voucher is deposited in the collection of the Laboratório de Zoologia Adriano Giorgi at the Universidade Federal do Pará, Campus de Altamira, Brazil.

**Paratopotypes.** Six adult males: LZAG 1382, 1383, 1385, 1386, 1389, 1391 and two adult females: LZAG 1384 and LZAG 1388 collected along with the holotype. Material deposited in the collection of the Laboratório de Zoologia Adriano Giorgi at the Universidade Federal do Pará, Campus de Altamira, Brazil.

**Paratypes.** Females LZAG 1188, 1190, 1191, 1197, BLM 1089 and a male LZAG 1387, collected by Suyanna Ferreira no Assurini, Travessão do Espelho (3˚19'40.11"S; 52˚10'13.82"O), Altamira municipality, Pará state. Material deposited in the collection of the Laboratório de Zoologia Adriano Giorgi at the Universidade Federal do Pará, Campus de Altamira, Brazil.

## Diagnosis

*Pristimantis giorgii* **sp. nov.** distinguishes from other species in the *P. conspicillatus* group by the following combination of characters (summarized in Table 2): (1) shagreen dorsal skin, with dorsal tubercles in the scapular region, supra ocular and laterally; lateral dorsal fold absent; discoidal fold present; smooth and granular belly skin texture laterally; (2) males with slit and vocal sac; (3) thigh surface with dark yellow spots on brown background; (4) four supernumerary tubercles present on palms of the hands and almost the same size as the subarticular tubercles projected; (5) side fringe along the fingers; (6) supernumerary tubercles present in the base of the foot's IV, conical and small format; (7) basal webbing and lateral fringe present between toes; (8) vocalization with three to four notes pulsed in length 0.017–0.074 s (N = 7);

## Comparison with other species

*Pristimantis giorgii* **sp. nov.** distinguishes from *P. latro* by the absence of lateral dorsal fold (present in *P. latro*) and vocalization composed of three to four notes (seven notes, see

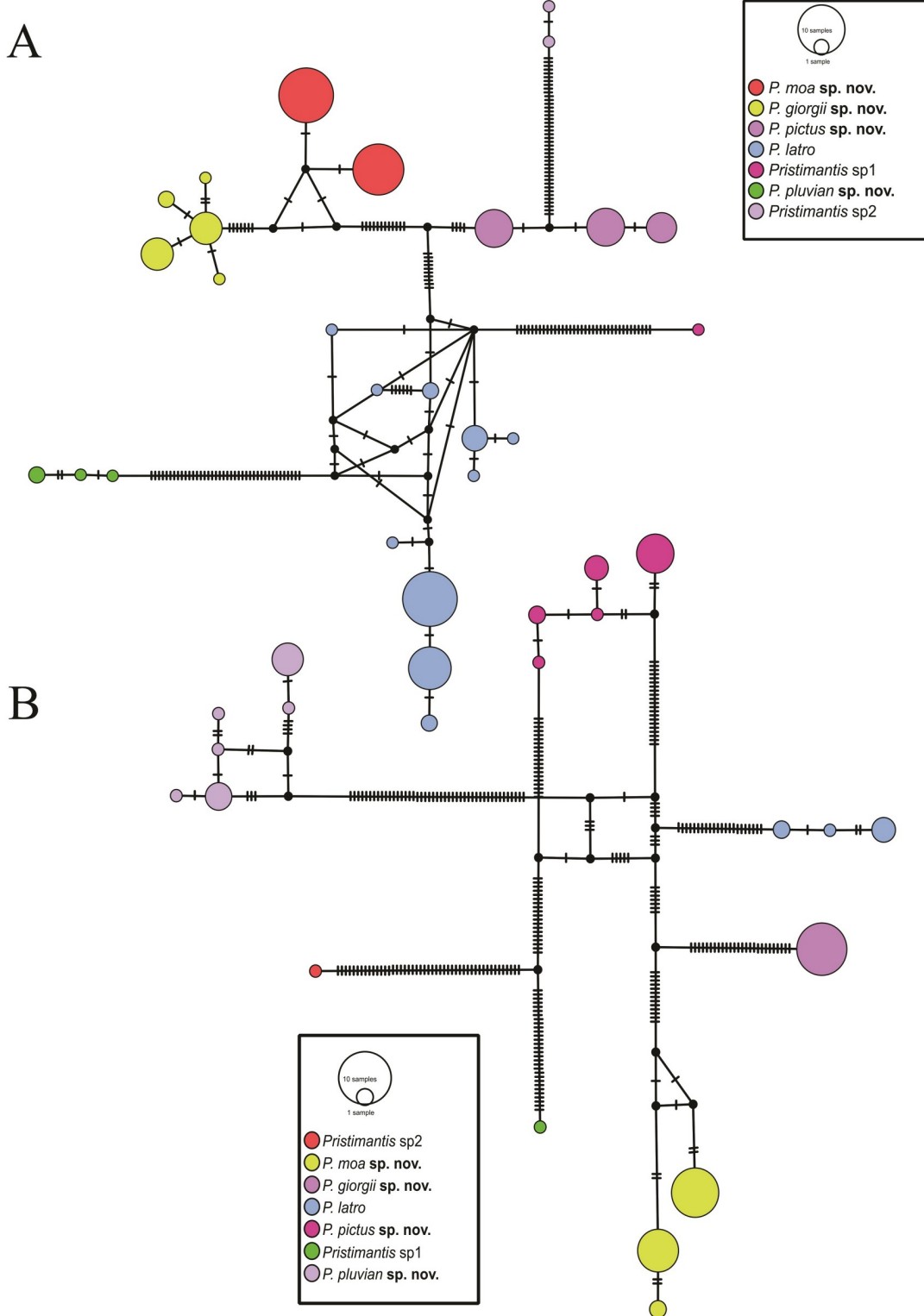

**Fig 3.** Median-joining haplotype network based on 16S (A) and COI (B). Each haplotype is represented by a circle, in which the area is proportional to its frequency. The traits indicate additional mutational steps of branches with more than one mutation. Black dots represent lost or unsampled haplotypes. Colours represent different species.

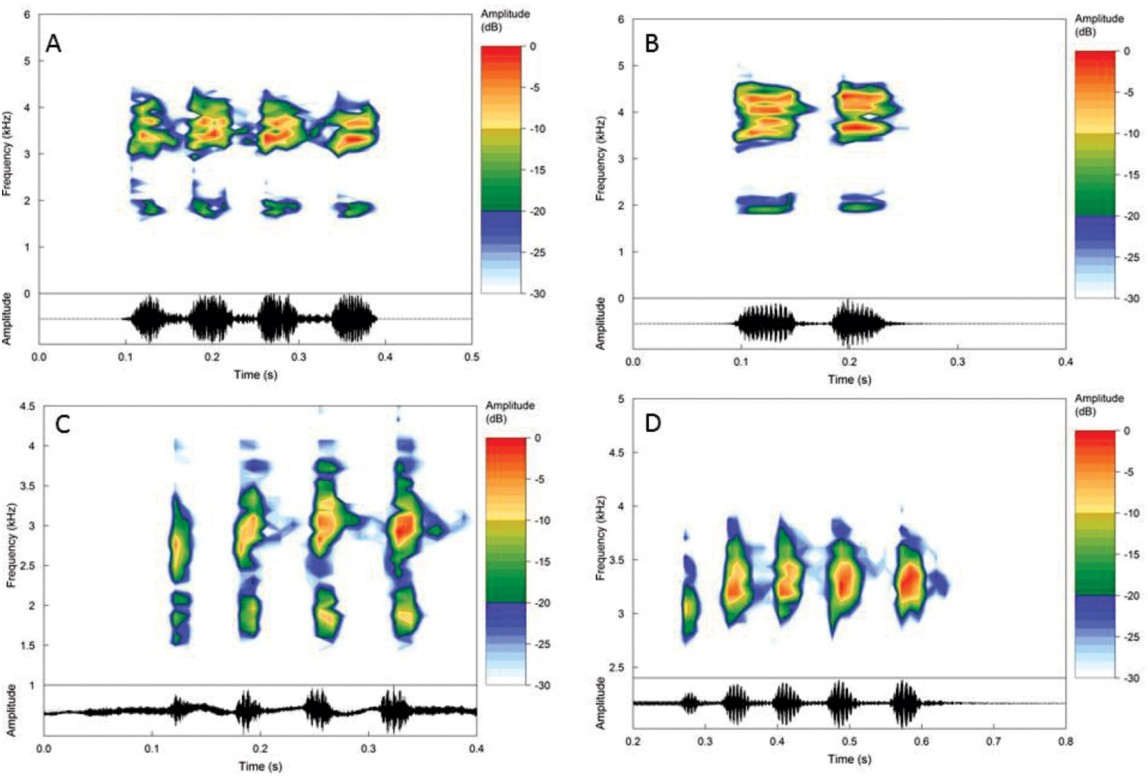

**Fig 4.** A) Vocalization of *Pristimantis giorgii* **sp. nov.** State of Pará; B) Vocalization of *P. pluvian* **sp. nov.** State of Mato Grosso; C) Vocalization of *P. pictus* **sp. nov.** Mato Grosso and Pará states and D) vocalization of *P. moa* **sp. nov.** Tocantins and Maranhão states.

Table 3); from *P. vilarsi* by presenting smooth belly texture in the center and granular laterally (smooth throughout the belly), presence of basal webbing in the feet (absent), vocalization of three to four notes (eight notes) [45], [46]; from *P. fenestratus* by presenting smooth belly texture in the center and granular laterally (smooth throughout the belly in *P. fenestratus*), fringe on toes prominent (weak) [31], [34], [47]; from *P. koehleri* by having a lateral fringe on the fingers (absent in *P. koehleri*), presence of basal webbing in the feet (absent), vocalization composed of three to four notes (four to eight notes) [14]; from *P. dundeei* by presenting a flat belly in the center and granular laterally (areolated in *P. dundeei*), presence of fringes on fingers (absent) [14], [48]; from *P. zeuctotylus* by presenting a divided palmar tubercle (undivided in *P. zeuctotylus*), white belly coloring with scattered black dots (black belly) [37]; from *P. chiastonotus* by presenting basal webbing and fringe on toes (absent in *P. chiastonotus*), tarsal fold present (absent); rostrum subacuminated in dorsal view (acuminate), dorsal tubercles present (absent), vocalization composed of three to four notes (one note) [37].

## Description of the holotype

Adult male 34.2 mm SVL. Dorsal skin shagreen, dorsal tubercles present, dorsolateral fold absent; smooth belly texture and granular laterally, granular posterior surface of thighs; head longer (39.5% of the SVL) than wide; snout subacuminated in dorsal view and curved antero-ventrally in lateral view; canthus rostralis weakly concave in dorsal view and in cross section,

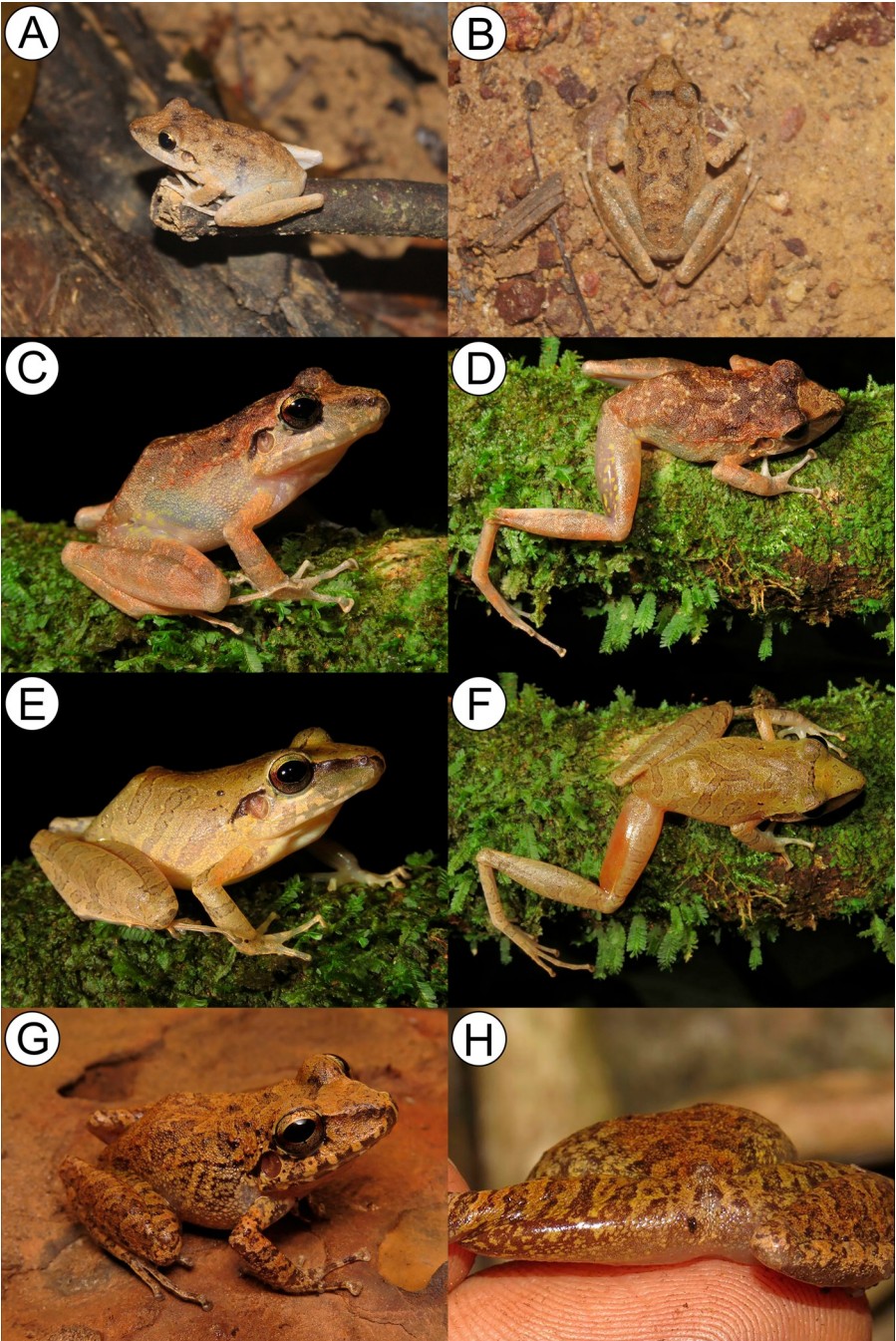

**Fig 5.** Color in life of: *Pristimantis giorgii* **sp. nov.** Pará state (A—D); *P. pictus* **sp. nov.** Mato Grosso state (E–H); *P. pluvian* **sp. nov.** Mato Grosso state (I–L) and *P. moa* **sp. nov.** Tocantins state (L–P).

flat loreal region; ovoid tongue, wider than long; dentigerous process of vomer oblique and posterior to choanae, eight visible teeth; eye 93% of distance from the eye to the nose; elliptical pupil; supraocular tubercles and inter-supraocular bar present; cranial crests absent; prominent supra tympanic fold, not contacting the eyelid; tympanic membrane 47.5% of EL, rounded and translucent, tympanic annulus prominent; relatively small hands, 26% of the SVL; relative length of fingers: II < IV < I < III; discs of fingers III and IV are wider than

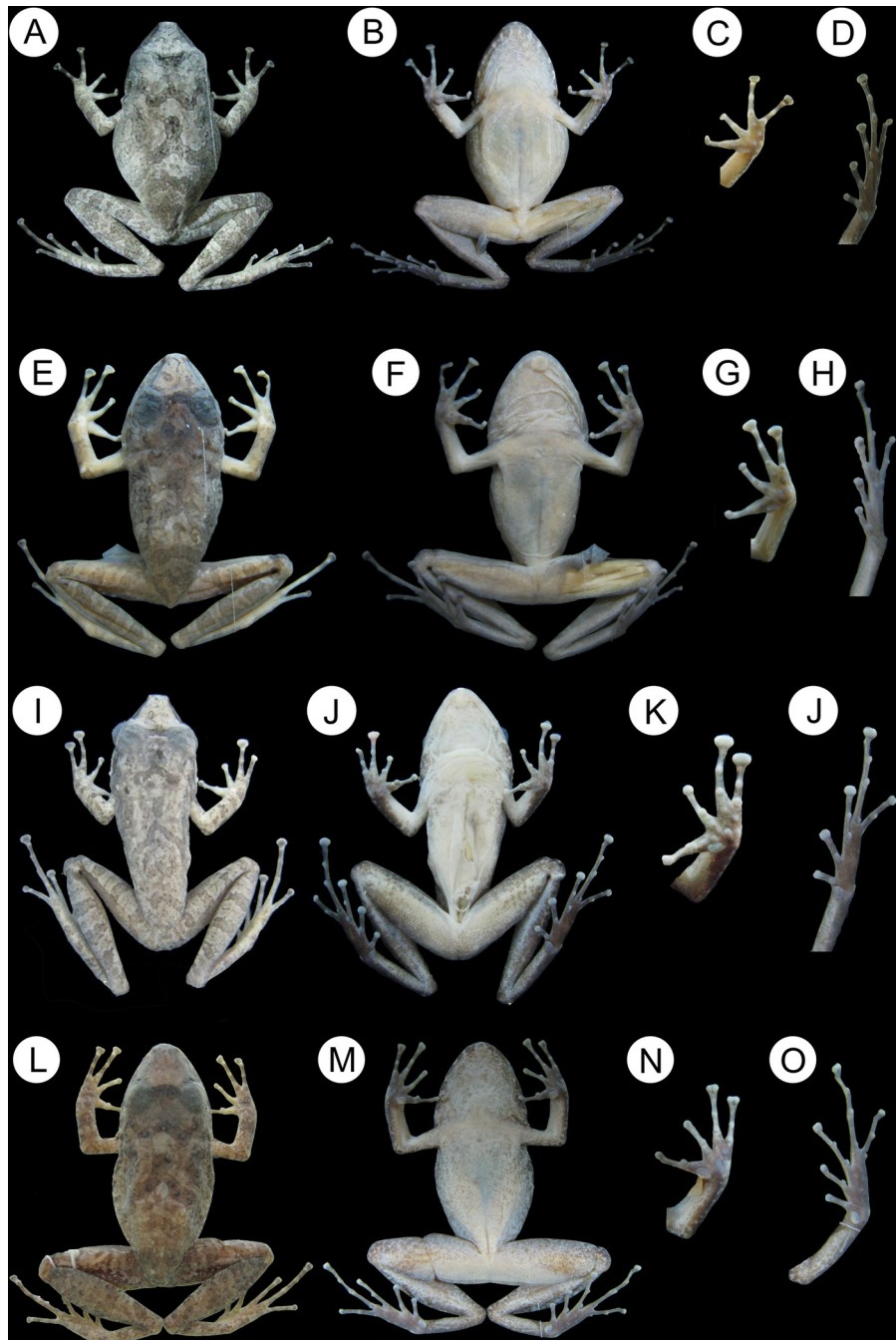

**Fig 6.** Holotype of *Pristimantis giorgii* **sp. nov.** (Portel, State of Pará): A) dorsal view (SVL = 34.2 mm), B) ventral view; C) left hand (8.9 mm) and D) left foot (16 mm). Holotype of *Pristimantis pictus* **sp. nov.** (Novo Mundo, State of Mato Grosso): E) dorsal view (SVL = 30.6 mm), F) ventral view; G) left hand (8.2 mm) and H) left foot (15.8 mm). Holotype of *Pristimantis pluvian* **sp. nov.** (Cotriguaçu, State of Mato Grosso): I) dorsal view (SVL = 28.7 mm), J) ventral view; K) right hand (7.3 mm) and L) right foot (14 mm). Holotype of *Pristimantis moa* **sp. nov.** (Palmeirante, Tocantins State): M) dorsal view (SVL = 32.9 mm), N) ventral view; O) left hand (8.8 mm) and P) right foot (15.9 mm).

fingers I and II; prominent, semi divided, heart-shaped external metacarpal tubercle; large internal palmar tubercle; one subarticular tubercle prominent on fingers I, II, III and IV; one supernumerary tubercles present at the base of fingers I, II, III and IV; long legs, tibia 56.1% of

**Table 2.** Comparison of the diagnostic characters among some species of the *Pristimantis conspicillatus* group, including new species: (1) belly texture (smooth, granular, granular posterolaterally), (2) dorsal tubercles (presence/absence of dorsal tubercles or short folds); (3) fringe on finger (prominent, weak, absent); (4) dorsolateral fold (prominent or weak); (5) fringe on toe (prominent, weak, absent); (6) basal membrane on toe (present or absent); (7) Inner tarsal fold (long, short or absent); (8) throat color pattern (stained, immaculate or light); (9) thigh surface coloring (yellow spots, reddish, spotted, strongly spotted, brown, other); (10) number of notes/call (1, 2, 3, 4, 5, more).

| Species | 1 | 2 | 3 | 4 | 5 | 6 | 7 | 8 | 9 | 10 |
|---|---|---|---|---|---|---|---|---|---|---|
| *P. fenestratus*\* | smooth | present | absent | - | weak | present | - | stained | other | 2 or 3 |
| *P. fenestratus*\*\* | smooth | absent | present | absent | weak | present | present | variable | - | 2 or 3 |
| *P. koehleri* | granular laterally | absent | absent | absent | weak | absent | present | light | brown | 3–8 |
| *P. dundeei* | granular | present | absent | absent | prominent | present | present | stained | other | 8 |
| *P. samaipatae* | smooth | absent | absent | absent | prominent | absent | present | stained | - | 1–3 |
| *P. zeuctotylus* | smooth | absent | absent | present | absent | absent | absent | stained | brown | |
| *P. chiastonotus* | smooth | absent | absent | present | absent | absent | absent | ligth | other | 1 |
| *P. vilarsi* | smooth | absent | absent | absent | absent | absent | present | immaculate | - | 8 |
| *P. latro* | smooth | present | present | present | weak | present | present | stained | brown | 7 |
| *P. giorgii* **sp. nov.** | **smooth and granular laterally** | **present or absent** | **present or absent** | **absent** | **proeminent** | **present** | **present** | **variable** | **spotted** | **3–4** |
| *P. pictus* **sp. nov.** | **smooth and granular laterally** | **present** | **present** | **present or absent** | **present** | **present** | **present** | **variable** | **yellow spots** | **4–5** |
| *P. pluvian* **sp. nov.** | **smooth and granular laterally** | **present or absent** | **present or absent** | **present or absent** | **present** | **Present** | **present** | **variable** | **reddish** | **2** |
| *P. moa* **sp. nov.** | **smooth and granular laterally** | **present** | **present** | **present or absent** | **present** | **present** | **present** | **variable** | **strongly spotted** | **3–5** |

Obs: (\*) Description by Duellman and Lehr (2009) for *Pristimantis fenestratus* from Peru; (\*\*) Description of Padial and De La Riva (2009) for *Pristimantis fenestratus* from Bolivia.

the SVL; relative length of toes: I <V <II <III <IV; well developed and oval inner metatarsal tubercle; external metatarsal tubercle much smaller than the internal; one subarticular tubercle on toes I and II; two subarticular tubercles on toes III and V; and three subarticular tubercles on toe IV; basal webbing and lateral fringes (weak) present on toes; inner tarsal fold present; outer tarsal tubercles absent.

## Measurements of holotype (in mm)

SVL: 34.2; HL: 13.5; HW: 12.7; SL: 6.7; DEN: 4.3; ID: 3.0; EL: 4.0; IoD: 2.8; EW: 3.5; TL: 1.9; AL: 7.6; HaL: 8.9; ThL: 15.5; TiL: 19.2; TaL: 9.7; FL: 16.0; LL: 24.6.

## Color in life

The dorsum is predominantly light brown with scattered dark brown bars. Posterior and anterior limbs with dark brown bars. Black labial bars. Black canthal stripe. White belly with tiny scattered black dots. Iris presents a golden coloration at the top and bottom, while in the anterior and posterior region is predominantly red. White groin area with few yellow spots. Posterior thigh surface predominantly light brown.

## Coloration in preservative

In alcohol, the coloration is predominantly brown in the dorsal region, both in males and females. The belly is white with tiny black dots scattered or black spots scattered all over the venter.

**Table 3. Diagnostic characters in vocalization among the species of *Pristimantis* used in this study.** Values are given as distribution in seconds (average ± SD). SD = Standard Deviation; Hz = Hertz; pop = population.

| Species | Notes/call | Call length (s) | Note length (s) | Pulse | Fundamental Frequency (Hz) | Dominant Frequency (Hz) | Notes | Calls | N° of individuals | N° of pop | Source |
|---------|-----------|-----------------|-----------------|-------|----------------------------|-------------------------|-------|-------|-------------------|-----------|--------|
| *P. fenestratus* | 2–4 (2.6 ± 0.6) | 0.157–0.458 (0.265 ±0.081) | 0.05–0.091 (0.063 ± 0.011) | 9–17 (12.9 ± 42.2) | 1542–2048 (1746 ± 158) | 1710–3591 (3086.3 ± 580.7) | 55 | 22 | 6 | 4 | Padial e De La Riva (2009) |
| *P. koehleri* | 3–8 (5.7 ± 1.0) | 0.173–0.644 (0.421 ± 0.159) | 0.002–0.054 (0.035 ± 0.006) | 5–9 (7.5 ± 1) | 1732–1971 (1853.5 ± 72.1) | 3245–3971 (3662.4 ± 128.9) | 119 | 21 | 6 | 2 | Padial e De La Riva (2009) |
| *P. samaipatae* | 1–3 (2 ± 0.2) | 0.082–0.106 (0.291 ± 0.168) | 0.059–0.141 (0.089 ± 0.016) | 11–23 (16.4 ± 2.6) | 1535–1834 (1704.9 ± 64.3) | 2922–3853 (3326.7 ± 175.9) | 160 | 98 | 12 | 4 | Padial e De La Riva (2009) |
| *P. vilarsi* | 8 | 0.521 | 0.001–0.002 | – | 1959–2256 | 3799–4444 | 8 | 1 | 1 | 1 | Heyer e Barrio-Amoros 2009 |
| *P. latro* | 7 | 0.402–0.581 (0.454 ± 0.068) | 0.031–0.045 (0.039 ± 0.005) | 6–9 (7.5 ± 2.12) | 1342–1448,6 (1381.41 ± 35.71) | 2635.89–3272 (3069.21 ± 253.61) | 49 | 7 | 6 | 2 | De Oliveira et al. 2017 |
| *P. giorgii* **sp. nov.** | 3–4 | 0.131–0.278 (0.171 ± 0.078) | 0.017–0.074 (0.04 ± 0.012) | 4–14 (8 ± 2.23) | 663.24–1872.28 (1511.69 ± 258.98) | 1660.17–4141.9 (3007.26 ± 511.25) | 42 | 13 | 7 | 1 | Present study |
| *P. pictus* **sp. nov.** | 4–5 | 0.216–0.302 (0.235 ± 0.021) | 0.017–0.033 (0.026 ± 0.004) | 3–12 (6.5 ± 2.06) | 1405.63–1914.71 (1676.63 ± 113.09) | 2487.42–3272.24 (2888.66 ± 159.59) | 56 | 15 | 4 | 1 | Present study |
| *P. pluvian* **sp. nov.** | 2 | 0.141–0.165 (0.152 ± 0.008) | 0.047–0.073 (0.056 ± 0.007) | 8–16 (11.8 ± 2.27) | 1617.75–1935.92 (1852.39 ± 92.12) | 3484.35–4311.59 (3645.42 ± 247.71) | 16 | 8 | 3 | 2 | Present study |
| *P. moa* **sp. nov.** | 3–5 (4.3 ± 0.6) | 0.212–0.380 (0.294 ± 0.048) | 0.023–0.064 (0.045 ± 0.009) | 4–18 (8.6 ± 2.85) | 1320.79–1660.17 (1509.64 ± 110.68) | 2657.11–3400.01 (3117.71 ± 158.74) | 75 | 20 | 6 | 2 | Present study |

## Variation

The texture of the venter is predominantly smooth in the center and granular laterally, but in some individuals it is completely smooth (BLM 1089, LZAG 1188, 1190, 1191, 1197, 1387, 1388). Dorsal tubercles absent in BLM 1089. All the individuals collected from the left bank of the lower Tocantins River present the dorsum and anterior and posterior limbs strongly barred, while those collected on the right bank of the middle Xingu River present a brown dorsum with few transversal bars. Females are larger than males (Average = 32.23, N = 25; Average = 32.09, N = 14)

## Etymology

The specific epithet is a patronymic to the Professor José Adriano Giorgi† of the Universidade Federal do Pará, due to his great contributions to knowledge about entomofauna of the Xingu region and since he accompanied us during the first collection of the species.

## Distribution, ecology and habitat

*Pristimantis giorgii* **sp. nov.** has been recorded in the interfluve Xingu/lower Tocantins in Pará state, Brazil throughout the municipalities of Altamira (Assurini, right bank of the middle Xingu river), Portel (left bank of the lower Tocantins River), Marabá and Melgaço (Flona de Caxiuanã) (Fig 1). This species can be found in preserved forest areas (Flona Caxiuanã and

Portel) or in some disturbed environments, *e.g.* forest fragments surrounded by pastures (agrovila do Assurini, Altamira municipality, Pará state). During the rainy season, this species becomes reproductively active, and males move up in the vegetation to call at a height of two meters; during the dry season, they can be found in the leaf litter.

*Pristimantis pictus* **sp. nov.** (Fig 5E–5H) urn:lsid:zoobank.org:act:AEE3D30E-203C-4215-BFA8-5EBFBEFD78FC

**Holotype.** (Fig 6E–6H), ABAM 2151, adult male, collected on 6 September 2014 in Novo Mundo, Mato Grosso State, Brazil (9˚28'1.68"S 55˚49'33.13"W) by Domingos de Jesus Rodrigues. Voucher is housed in the Acervo Biológico da Amazônia Meridional (acronym ABAM) at the Universidade Federal do Mato Grosso, Campus de Sinop, Brazil.

**Paratopotypes.** 11 females: ABAM 1482, 1602, 1829–31, 2146, 2129, 2150, 2155, 2154, 2161. One male (ABAM 2148) and one juvenile (ABAM 2241). Vouchers are housed in the Acervo Biológico da Amazônia Meridional at the Universidade Federal do Mato Grosso (UFMT), Campus de Sinop, Brazil.

**Paratypes.** Two females: ABAM 1500 and ABAM 1525 from Cotriguaçu, Mato Grosso state. Two females: ABAM 2109–10, collected in Tabaporã, Mato Grosso state. Three males ABAM 3077, 3078 and 3081 collected in Sinop, Mato Grosso state. Material deposited in the biological collection of the Amazônia Meridional da Universidade Federal do Mato Grosso, Campus de Sinop, Brazil. Three males (ZUFMS-AMP 8538–40), two females (ZUFMS-AMP 8541 and 8544) and two juveniles (ZUFMS-AMP 8542–43) collected in Jacareacanga, Pará state, Brazil. Material deposited in the zoological collection of the Universidade Federal do Mato Grosso do Sul (acronym ZUFMS).

## Diagnosis

*Pristimantis pictus* **sp. nov.** differs from the other species in the *P. conspicillatus* group by the following character combinations (summarized in Table 2): (1) dorsal skin shagreen, dorsal tubercles present distributed mainly in the lateral dorsal region, supra-ocular tubercles absent, smooth belly texture in the central part and granular laterally; (2) males with vocal slit and vocal sac present; (3) four supernumerary tubercles on the palm of the hands almost the same size as the subarticular, conical shaped; (4) lateral fringe along the fingers of the hands; (5) basal webbing and lateral fringe between toes; (6) vocalization with four to five notes with length ranging from 0.017–0.033 s (N = 4) and pulsed; and (7) posterior surface of the thighs with light yellow patches on brown background, white groin with yellow spots.

## Comparison with other species

*Pristimantis pictus* **sp. nov.** distinguishes from *P. giorgii* **sp. nov.** by presenting yellow spots on a brown background on the posterior thigh surface (light yellow spots absent in *P. giorgii* **sp. nov.**), occurs in northern Mato Grosso state (eastern Pará state), vocalization composed of four to five notes (three to four notes); from *P. latro* by vocalization composed of four to five notes (consisting of seven notes in *P. latro*), occurs in northern Mato Grosso state (Tapajós/lower Tocantins interfluve); from *P. vilarsi* by presenting smooth belly texture in the central and granular side laterally (smooth in *P. vilarsi*), presence of basal webbing in toes (absent), vocalization composed of four to five notes (composed of eight) [45], [46]; from *P. fenestratus* by presenting smooth belly texture in the central and granular side laterally (smooth in *P. fenestratus*), presence of light yellow spots on brown background on the posterior thigh surface (light yellow spots absent) [14], [31], [47]; from *P. koehleri* by the smooth belly texture in the central and granular side laterally (finely granular in *P. koehleri*), vocalization consisting of four and five notes (from four to eight notes) [14]; from *P. samaipatae* by possessing white

belly with tiny black dots scattered randomly (immaculate in *P. samaipatae*), semi-divided external palmar tubercle (divided) [14], [49]; from *P. dundeei* by possessing smooth belly texture in the center and granular laterally (areolate in *P. dundeei*), presence of fringe on the fingers (absent) [14], [48]; from *P. zeuctotylus* by presenting semi-divided palmar tubercle (whole in *P. zeuctotylus*), white belly with tiny black dots scattered randomly (black belly) [37]; from *P. chiastonotus* by presenting basal webbing and fringe on toes (absent in *P. chiastonotus*), inner tarsal fold present (absent), dorsal tubercle present (absent), vocalization composed of four to five notes (one note) [37].

## Description of the holotype

Adult male 30.6 mm SVL. Dorsal skin strongly shagreen, dorsal tubercles present, mainly in the lateral region of the dorsum, dorsolateral fold present or absent; smooth belly texture and granular laterally, granular posterior surface of thighs with yellow spots on brown background; head longer (38.2% of the SVL) than wide; long snout (49.5% do HL), subacuminate in dorsal view and protruding in lateral view; canthus rostralis slightly concave in dorsal view and in cross section, flat loreal region; ovoid tongue, wider than long; dentigerous process of vomer oblique and posterior to choanae; the distance from the eye to the nose is the same horizontal diameter of the eye; elliptical pupil; supraocular tubercles present; cranial crests absent; prominent supra tympanic fold, not contacting the eyelid; tympanic membrane 41% of EL, rounded and translucent tympanic annulus prominent; relatively small hands, 28.7% of the SVL; relative length of fingers: II < IV < I < III; discs of fingers III and IV are wider than fingers I and II; prominent, semi divided, heart-shaped external metacarpal tubercle; large internal palmar tubercle; one subarticular tubercle prominent on fingers I and II, two prominent subarticular tubercles on fingers III and IV; one supernumerary tubercles present at the base of fingers I, II, III and IV almost the same size as subarticular tubercles; long legs, tibia 59.1% of the SVL; relative length of toes: I <II <V <III <IV; well developed and oval inner metatarsal tubercle; external metatarsal tubercle much smaller than the internal one; one subarticular tubercle on toes I and II; two subarticular tubercles on toes III and V; and three subarticular tubercles on toe IV; basal webbing and lateral fringes present on toes, absent supernumerary tubercle; long inner tarsal fold present.

## Measurements of holotype (in mm)

SVL: 30.6; HL: 11.7; HW: 11.3; SL: 5.8; DEN: 3.9; ID: 2.4; EL: 3.9; IoD: 2.6; EW: 3.0; TL: 1.6; AL: 6.3; HaL: 8.8; ThL: 15.6; TiL: 18.1; TaL: 8.0; FL: 15.2; LL: 23.0.

## Color in life

Reddish dorsum with yellow dots on the flanks forming a division between the lateral and the whitish venter. Posterior and anterior limbs present dark brown bars. Evident labial bar. Canthal stripe weak,. White belly with tiny black dots scattered randomly. Iris features a golden coloration at the top and bottom, whereas in the anterior and posterior region it is predominantly red. The posterior surface of the thigh presents bright yellow spots on a brown background. Groin area with yellow spots, extending across the lateral half of the body.

## Coloration in preservative

Coloration is predominantly brown in dorsal view, both in females and males. The belly can be white with tiny black dots or black spots randomly distributed. The rostral band present in

some individuals appears more erased, hindering its visualization. The yellow spots on the posterior surface of the thigh are whitish.

## Variation

The texture of the dorsal surface is usually strongly shagreen (ABAM 1829, 1830, 2110, 2129, 2146, 2148, 2150, 2151, 2154, 2155, 2161, 2241, 3077, 3078, 3081), but is shagreen in the individuals: ABAM 1482, 1500, 1525, 1602, 1831, 2109, ZUFMS-AMP 8538–44. An adult female (ZUFMS-AMP 8544) does not present dorsal tubercles. The lateral dorsal folds are most common, found in most of the type series (ABAM 1500, 1525, 1829, 1830, 1831, 2110, 2129, 2146, 2148, 2150, 2151, 2154, 2155, 3078, ZUFMS-AMP 8538, 8540, 8541 and 8543), absent only in: ABAM 1482, 1602, 2109, 2161, 2241, 3077, 3081, ZUFMS-AMP 8539, 8542 and 8544. The texture of the belly is predominantly smooth in the center and granular laterally, however, some individuals have a completely smooth belly (ABAM 1525 and ZUFMS-AMP 8542), the two individuals are juvenile, and it may be difficult to identify the granules laterally. The venter is heavily stained in individuals ABAM 1830, 1831, 2154 and 2155. The dorsal coloration may be more commonly present with transverse and interorbital bars, but some individuals present distinct patterns (ABAM 3081, ZUFMS-AMP 8543–44). Females are slightly larger than males (Average = 33.84, N = 17; Average = 32.95, N = 5)

## Etymology

The specific epithet "*pictus*" comes from the Latin (*pictus* = painted, stained) and refers to the yellow dots that are characteristic to individuals of this species, facilitating its identification in relation to other *Pristimantis* species in its occurrence region.

## Distribution, ecology and habitat

*Pristimantis pictus* **sp. nov.** has been registered in the municipalities of Cotriguaçu, Novo Mundo, Tabaporã, Paranaíta and Sinop in Mato Grosso state and Jacareacanga in Pará state, Brazil (Fig 1). It can be found at the edges of preserved forests, close to pastures.

*Pristimantis pluvian* **sp. nov.** (Fig 5I–5L) urn:lsid:zoobank.org:act:C6D52DAF-C3A6-487E-8BC8-D2376832D1BD

**Holotype.** (Fig 6I–6L), ABAM 3221, adult male, collected on 25 January 2018 in the Ilha Juruena, from the Cotriguaçu municipality, Mato Grosso state, Brazil (9°55'13.32"S 58°14'29.45"W) by Domingos de Jesus Domingues. Voucher is housed in the Acervo Biológico da Amazônia Meridional (acronym ABAM) at the Universidade Federal do Mato Grosso (UFMT), Campus de Sinop, Brazil.

**Paratopotypes.** 13 females, ABAM 255, 305, 530, 531, 545, 549, 664, 739, 768, 794, 1555, 1556, 2173. Six males (ABAM 344, 345, 394, 393, 427, 1658) and 10 juveniles (ABAM 550, 734, 735, 740, 741, 742, 770, 781, 2233, 2234). Collected in the same locality as the holotype. Vouchers are housed in the Acervo Biológico da Amazônia Meridional at the Universidade Federal do Mato Grosso, Campus de Sinop, Brazil.

**Paratypes.** One female, ABAM 3040, collected from the Ipiranga do Norte municipality, Mato Grosso state. Material deposited in the Acervo Biológico da Amazônia Meridional at the Universidade Federal do Mato Grosso, Campus de Sinop, Brazil. Eight females: ZUFMS-AMP 8546, 8547, 8578, 8579, 8580, 8582, 8583, 8586; a male (ZUFMS-AMP 8584), and a juvenile (ZUFMS-AMP 8585) collected in the Paranaíta municipality, Mato Grosso state, deposited in the Coleção Zoológica da Universidade Federal do Mato Grosso do Sul.

## Diagnosis

*Pristimantis pluvian* **sp. nov.** differs from the other species in the group by the following combination of characters (summarized in Table 2): (1) dorsal skin smooth or shagreen, lateral dorsal fold present, dorsal tubercles present mainly in the lateral part of the body and few in the scapular region, smooth and granular belly texture laterally; (2) males with slits and vocal sac present; (3) light venter with tiny black dots scattered; (4) four or five supernumerary tubercles in the palm of the hands almost the same size as the subarticular and conical; (5) basal webbing and fringe between toes; (6) one to three small supernumerary tubers present in the foot; (7) vocalization consisting of two notes with length ranging from 0.047–0.073 s (N = 3) and pulsed; (8) posterior surface of the thighs reddish and without spots.

## Comparison with other species

*Pristimantis pluvian* **sp. nov.** distinguishes from *P. giorgii* **sp. nov.** by presenting vocalization consisting of two notes (from three to four notes in *P. giorgii* **sp. nov.**), lateral dorsal fold present (absent); from *P. pictus* **sp. nov.** by the presence of the posterior thigh surface reddish (light yellow patches in *P. pictus* **sp. nov.**), vocalization consisting of two notes (from four to five notes); from *P. latro* by vocalization composed of two notes (seven notes), northern Mato Grosso state (Tapajós/Xingu interfluve); from *P. vilarsi* by presenting smooth belly texture in the center and granular laterally (smooth along the whole belly in *P. vilarsi*), presence of basal webbing in the feet (absent), vocalization consisting of two notes (eight notes) [45], [46]; from *P. fenestratus* by presenting smooth belly texture in the center and granular laterally (smooth in *P. fenestratus*), vocalization consisting of two notes (from three to five notes), northern Mato Grosso state (western Amazonia) [31], [47]; from *P. koehleri* by the texture of the flat belly in the center and granular laterally (finely granular along the belly in *P. koehleri*), vocalization consisting of two notes (from four to eight notes) [14]; from *P. samaipatae* by possessing white belly with black dots scattered randomly (immaculate in *P. samaipatae*), external palmar tubercle semi-divided (divided) [14]; [48]; from *P. dundeei* by possessing smooth belly texture in the center and granular laterally (areolate along the belly in *P. dundeei*), lateral dorsal fold present (absent) [14], [48]; from *P. zeuctotylus* by a semi-divided palmar tubercle (undivided in *P. zeuctotylus*), white belly with black dots scattered randomly (dark belly) [37]; from *P. chiastonotus* for presenting basal webbing and fringe on toes (absent in *P. chiastonotus*), tarsal fold present (absent), dorsal tubercles present (absent), vocalization consisting of two notes (one note) [37].

## Description of the holotype

Adult male 28.7 mm SVL. Dorsal skin shagreen, dorsal tubercles present, mainly on the side of the body and dorsal lateral fold present; texture of the belly smooth and granular laterally, granular posterior surface of thighs; head longer (34.1% of the SVL) than wide; long snout (54.1% do HL), truncate in dorsal view and long and protruding in lateral view; *canthus rostralis* slightly concave in dorsal view and in cross section,, flat loreal region, obvious facial mask, weak labial bars; ovoid tongue, wider than long; dentigerous process of vomer oblique and posterior to choanae; distance from the eye to the nose (3.7 mm) is greater than the horizontal diameter of the eye (3.4 mm); elliptical pupil; absent supraocular tubercles; absent cranial crests; prominent supra tympanic fold, contacting the eye; tympanic membrane 41.2% of EL, rounded and translucent, tympanic annulus prominent; relatively small hands, 25.4% of the SVL; relative length of fingers: II < IV < I < III; discs of fingers III and IV are wider than fingers I and II; prominent, semi divided, heart-shaped external metacarpal tubercle; large internal palmar tubercle in contact with nuptial pad; one subarticular tubercle prominent on

fingers I and II, two prominent subarticular tubercles on fingers III and IV; supernumerary tubercles present at the base of fingers I, II, III and IV and above the inner palmar tubercle; long legs, tibia 56.4% of the SVL; relative length of toes: I <V <II <III <IV; well developed and oval inner metatarsal tubercle; external metatarsal tubercle much smaller than the internal; one subarticular tubercle on toes I and II; two subarticular tubercles on toes III and V; and three subarticular tubercles on toe IV; one to three plantar supernumerary tubercle; basal webbing and lateral fringes present on toes; inner tarsal fold present and long.

## Measurements of holotype (in mm)

SVL: 28.7; HL: 9.8; HW: 9.3; SL: 5.3; DEN: 3.7; ID: 2.3; EL: 3.4; IoD: 2.3; EW: 2.6; TL: 1.4; AL: 5.8; HaL: 7.3; ThL: 14.5; TiL: 16.2; TaL: 7.9; FL: 14.0; LL: 21.6.

## Color in life

Dorsum can be reddish or greenish with transversal bars and dark brown interorbital. Posterior and anterior limbs have brown bars. Black labial bars. Facial mask weak Canthal stripe evident. White belly. Iris presents a golden coloration on top and bottom, while the anterior and posterior regions are predominantly red. Posterior surface of the reddish thigh without stains. Reddish groin region.

## Coloration in preservative

The coloration is predominantly brown in the dorsal region, both in males and females. The belly can be white with tiny black or immaculate dots. The transversal bands present in some individuals appear more erased and in others may be difficult to observe.

## Variation

The female ABAM 545, the male ABAM 427 and the juvenile ABAM 735 and 781 present the smooth belly, while all others in this type series present the smooth belly in the center and granular laterally. In ABAM 305 and 781 the dorsum is smooth, while the shagreen condition is present in the other members of the type series, strongly shagreen in ABAM 2173, 2233, 2234 and ZUFMS-AMP 8547. The dorsal tubercles are common, absent only in the individuals ABAM 427, 781, 3040, ZUFMS-AMP 8579, 8582 and 8584. The dorsolateral fold is absent in some individuals (ABAM 344, 345, 545, 549, 550, 740, 741, 742, 781, 1555, 1556, 2173, ZUFMS-AMP 8582, 8583, 8586) of the type series. In juveniles ABAM 2233 and 2234 it was not possible to observe discoidal fold due to the conservation status of the specimens. Individuals ABAM 305, 345, 344, 393, 427, 1555 have no fringe along the fingers. ABAM 305, ZUFMS-AMP 8583, 8585, 8586 present the immaculate venter, while the rest of the type series present spots or black spots scattered randomly. Individuals ABAM 255, 345, 770, 2173 and ZUFMS-AMP 8586 have a white longitudinal band from the tip of the rostrum to the cloaca. Females larger than males (Average = 32.3, N = 23; Average = 32.01, N = 14)

## Etymology

The specific epithet "*pluvian*" comes from Latin and means rain. The frogs of this genus are known as Amazon Rain Frogs.

## Distribution, ecology and habitat

*Pristimantis pluvian* **sp. nov.** is registered in the municipalities of Cotriguaçu, Ipiranga do Norte and Paranaíta in Mato Grosso state, Brazil (Fig 1). It can be found in conserved areas of

forests or areas with some environmental disturbances, *e.g.*, forest fragments surrounded by pastures. During the reproductive period, males rise in vegetation and vocalization. During the dry period they are predominantly on the ground.

*Pristimantis moa* **sp. nov.** (Fig 5M–5P) urn:lsid:zoobank.org:act:D458DB33-6D85-4C80-963A-2AFA02011977

**Holotype.** (Fig 6M–6P), ZUFMS-AMP 8558, adult male, collected on 17 September 2016 in Palmeirante municipality, Tocantins state, Brazil (07˚51'35.83"S; 47˚57'12.38"W) by Baeta, A. P.

**Paratopotypes.** One adult female (ZUFMS-AMP 8557) collected in the same locality and date of holotype by Baeta, A.P. Voucher is housed in the Coleção Zoológica da Universidade Federal do Mato Grosso do Sul, Brazil.

**Paratypes.** Five female adults: ZUFMS-AMP 8548, 8550, 8575, 8576, 8577 and three juveniles: ZUFMS-AMP 8549, ZUFMS-AMP 8551 and ZUFMS-AMP 8574, collected from Araguaína and ZUFMS-AMP 8587–8594 in Riachão, Maranhão state. Vouchers are housed in the Coleção Zoológica da Universidade Federal do Mato Grosso do Sul, Brazil.

## Diagnosis

*Pristimantis moa* **sp. nov.** differs from other species in the *P. conspicillatus* group by the following combination of characters (summarized in the Table 2): (1) shagreened dorsal skin, dorsal tubercles present mainly in the scapular region, discoidal fold present, smooth belly skin texture in the center and granular laterally; (2) ventral coloration with tiny dots or black spots on white background; (3) a subarticular tubercle in fingers I and II, and two in fingers III and IV, all are large and conical; (3) posterior surface of the thigh strongly stained dark yellow on brown background;(4) supernumerary tubercles presented at the base of the finger I, II, III and IV, almost the same size as the subarticular tubercles; (5) side fringe along the fingers of the hand; (6) supernumerary tubercle present at the base of toes III and IV, smaller than subarticulars; (7) basal webbing well developed and lateral fringe present between toes; (8) vocalization consisting of three to five notes, notes are on average 45 ms long (minimum 23.8 and maximum 64.6, ± 9.1); (9) occurs in the right margin of the Araguaia River and left and right margins of the Tocantins River.

## Comparison with other species

In parentheses, the character state of the nominal species compared to the new species. *Pristimantis moa* **sp. nov.** distinguishes from *P. giorgii* **sp. nov.** by presenting vocalization composed of three to five notes (three to four in *P. giorgii* **sp. nov.**), posterior surface of the thigh strongly stained yellow on a dark background (weakly stained yellow on a dark background), it occurs on the right bank of the Araguaia river and left and right banks of the Tocantins river (left bank of the lower Tocantins river); from *P. pictus* **sp. nov.** by presenting vocalization composed of three to five notes (four to five in *P. pictus* **sp. nov.**), posterior surface of the thigh strongly stained yellow on a dark background (yellow spots on brown background); from *P. pluvian* **sp. nov.** by posterior surface of the thigh strongly stained yellow on a dark background (reddish in *P. pluvian* **sp. nov.**), vocalization composed of three to five notes (two notes); from *P. latro* by vocalization composed of three to five notes (seven notes in *P. latro*), posterior surface of the thigh strongly stained yellow on a dark background (almost spotless); from *P. vilarsi* by presenting the smooth belly texture in the center and granular laterally (smooth in *P. vilarsi*), presence of basal webbing in toes (absent), vocalization with three to five notes (eight notes) [45], [46]; from *P. fenestratus* by presenting the smooth belly texture in the center and granular laterally (smooth in *P. fenestratus*), vocalization consisting of three to five notes (from

two to three notes), occurs in the eastern Brazilian Amazon, Tocantins state (occurs in the western Amazon) [14], [31], [47]; from *P. koehleri* by presenting the smooth belly texture in the center and granular laterally (finely granular in *P. koehleri*), vocalization composed of three to five notes (four to eight notes), shorter vocalization 0.294s (0.421s) [14]; from *P. samaipatae* by having a white belly with randomly scattered black dots (immaculate in *P. samaipatae*), semi-divided external palmar tubercle (split) [14], [48]; from *P. dundeei* by presenting the smooth belly texture in the center and granular laterally (areolate in *P. dundeei*) [14], [48]; from *P. zeuctotylus* by a divided palmar tubercle (entire in *P. zeuctotylus*), a white belly with randomly scattered black dots (black belly) [37]; from *P. chiastonotus* by presenting basal webbing and fringe on the toes (absent in *P. chiastonotus*), tarsal fold present (absent), dorsal tubercles present (absent), vocalization composed of three to five notes (one note) [37].

## Description of the holotype

Adult male 32.9 mm SVL. Dorsal skin shagreened, presence of dorsal tubercles and dorsal lateral fold, five tubercle in the scapular region, with the dark base forming a "W", interorbital bar; texture of the belly smooth and granular laterally, granular anterior surface of thighs; head longer (38.6% of the SVL) than wide; long snout, truncate in dorsal view and long and protruding in lateral view; convex canthus rostralis, flat loreal region; dentigerous process of vomer oblique and posterior to choanae, longer than wide tongue; eye diameter (4.4 mm) greater than distance from eye to nostril (4.0 mm); elliptical pupil; supraocular tubercles absent; cranial crests absent; prominent supra tympanic fold with tubercle present in the posterior region, not contacting the eye; tympanic membrane 47.7% of EL, rounded and translucent, tympanic annulus prominent; relatively small hands, 26.7% of the SVL; relative length of fingers: II < IV < I < III; discs of fingers III and IV are wider than fingers I and II; prominent, semi divided, heart-shaped external metacarpal tubercle; large internal palmar tubercle; one subarticular tubercle prominent on fingers I and II, two prominent subarticular tubercles on fingers III and IV; supernumerary tubercles present at the base of fingers I, II and III; long legs, tibia 57.4% of the SVL; relative length of toes: I <II <V <III <IV; well developed and oval inner metatarsal tubercle; external metatarsal tubercle much smaller than the internal one; one subarticular tubercle on toes I and II; two subarticular tubercles on toes III and V; and three subarticular tubercles on toe IV, all large and conical; plantar supernumerary tubercle present at the base of the toe IV and smaller than the subarticular; basal webbing well developed and lateral fringes present on toes (weak); inner tarsal fold present and long.

## Measurements of holotype (in mm)

SVL: 32.9; HL: 12.7; HW: 14.4; SL: 6.7; DEN: 4; ID: 2.6; EL: 4.4; IoD: 2.7; EW: 3.3; TL: 2.1; AL: 7.3; HaL: 8.8; ThL: 16.7; TiL: 18.9; TaL: 8.6; FL: 15.9; LL: 24.

## Color in life

Brown dorsum with darker transversal bars. Posterior and anterior limbs present dark brown bars. Black lip bars. Canthal stripe weak, almost absent in some individuals. White belly with tiny dots or dark spots. The iris has golden coloration on top and bottom, while in the anterior and posterior region it is predominantly red.

## Coloration in preservative

The coloration is predominantly brown on the back, in both males and females. The belly can present tiny black dots or black spots scattered randomly.

## Variation

The texture of the belly in ZUFMS-AMP 8575 appears almost completely granular and smooth only in the anterior portion of the belly, while the ZUFMS-AMP 8552, which is a juvenile, presents a smooth belly and may be difficult to observe granules. Individuals ZUFMS-AMP 8551, 8557, 8574, 8577 do not present dorsolateral fold, present in the rest of the type series. Individuals ZUFMS-AMP 8575 and ZUFMS-8550 present a face mask more evident than other individuals, making it difficult to visualize the canthal stripe. The side folds are present in the dorsal part of the series type (ZUFMS-AMP 8548/50, ZUFMS-AMP 8558, 8575/76) and absent in the others. Females are larger than males (Average = 31.7, N = 7; Average = 28.5, N = 9)

## Etymology

The specific epithet is a patronym of the Capoeira Master Romualdo Rosário da Costa†, known as Mestre Moa do Katendê, for his struggles for the black movement of Bahia.

## Distribution, ecology and habitat

*Pristimantis moa* **sp. nov.** is registered in the municipalities of Palmas, Palmeirante and Araguaína in Tocantins state and Carolina, Balsas and Riachão in Maranhão state, Brazil (Fig 1). It can be found in conserved forest areas or in areas with some environmental disturbances, *e.g.*, forest fragments surrounded by pastures. During the reproductive period, males rise in vegetation and vocalization. During the dry period they are predominantly on the ground.

## Discussion

Currently, only two species of the *Pristimantis conspicillatus* group are known in the eastern Brazilian Amazon, in the southern Amazon River, *P. latro* (Pará state) and *P. dundeei* (northern Mato Grosso state). With the four new species described in this work, this number increases to six, representing a 200% increase in the number of species for this region. Likewise, this represents a 12% increase in the diversity of species of the *P. conspicillatus* group, leading the numbers from 34 to 38 species in total, of which 20 right now are occupying the Brazilian territory. The fact of having discovered four new species of *Pristimantis* exploring just 11.5% of the total area of the distribution of this group, suggests the existence of a greater diversity still underestimated and highly promising in that remaining 88.5% of territory inhabited by the species of the group along the Neotropic; even when the group currently inhabits the Andes and Amazon, the worldwide regions with the principals ecological and hydro-geomorphological process causing diversification [50–54]. This study confirms that in the specious *Pristimantis conspicillatus* group, apparently a lot of efforts on integrative taxonomic research is needed for studies on new species, cryptic diversity, speciation processes and biogeography, due to their great capacity and varied mechanisms of diversification [55]. In the Neotropical region, estimates based on approaches integrating different character types (e.g. morphology, molecular and bioacoustics) showed increased frog richness between 150 and 300% [49]. Studies indicate that in the last 250 years of taxonomic classification only 14% of all terrestrial species have been described [56]. Our results are part of a broad set of evidence that indicates that the Neotropical biodiversity, particularly through the Amazon, cannot be trivialized [57–62].

In the last decade, 155 new species of *Pristimantis* have been described for South America, mainly from the Andean regions of Peru, Colombia, Ecuador and the Guianas Shield. The heterogeneity in the topography of the Andes, accompanied by the life history of *Pristimantis*

(direct development, high endemism and ability to colonize various habitats, including high altitudes) could be partly responsible for this notorious biodiversity [4]. The regions with the most homogeneous topography (Amazon Basin) present a high number of species that undergo a process called morphological stasis [16], where species that suffer strong selection by the environment or physiological characters for adaptation in a specific habitat, may not present morphological changes between species [63]. This morphological conservatism in *Pristimantis* represents one of the main problems for accurate identification of the species, resulting in an underestimated number of species for this genus [64], [65].

The occurrence of sympatric *Pristimantis* species has been recorded in several studies [45], [66], [67]. In this study, *P. giorgii/P. latro* and *P. pictus/P. pluvian* can be found in sympatry, with subtle morphological differences, although with great variation in genetic distance and vocalization (see Tables 1 and 3). Many cryptic species complexes are sympatric and not necessarily sister species, providing strong indirect evidence that represent independent haplogroups and cannot hybridize [68], [69]. The current sympatric distribution may be the result of secondary contact after the expansion of past populations [70]. The great variation observed in the vocalization of sympatric species may be the result of cryptic diversification and acoustic radiation, making it potentially evident in highly diversified and widely distributed anuran groups, as the *Pristimantis* genus [48]. Phylogenetic reconstruction shows that *P. pluvian*, *P. fenestratus*, *P.* sp. (municipality of Juruti, PA) and *P.* sp. (municipality of Borba, AM) belong to a hard polytomy, which may be common in organisms experiencing rapid speciation but with imcomplete lineage sorting in mtDNA. On the other hand, this may be because we do not have enough molecular data to describe how these lineages are related.

The concern with the global decline of amphibian species is an old debate in the academic community [70–73]. One of the main causes of this phenomenon is the loss of habitats through deforestation and fragmentation [74–76]. Once many species have not yet been described, efforts to catalog and describe biodiversity need to be prioritized [16], as in the case of *Pristimantis* sp1, that occurs only in the area called Terra do Meio, deserving greater attention for its taxonomic resolution, incorporating morphological and molecular data, in order to help in the knowledge of the biodiversity of the eastern Brazilian Amazon and the conservation of the species.

## Supporting information

**S1 File. PCR details and sequencing reaction protocol.**
(PDF)

**S2 File. Specimens examined for molecular analysis.**
(PDF)

**S3 File. Morphometric measurements of *Pristimantis* in this study.**
(PDF)

## Acknowledgments

We thank Evonildo Gonçalves from the Universidade Federal do Pará (Campus de Belém) for assisting in molecular protocols at Instituto Evandro Chagas, as well as the entire staff at the Tecnologia Bio-molecular laboratory who helped obtain genetic data; Gabriel Iketani from the Universidade Federal do Oeste do Pará by help in sequencing; the Coleção de Anfíbios e Répteis of the Instituto Nacional de Pesquisas da Amazônia (INPA-H) and the Coleção Herpetológica of the Museu Paraense Emílio Goeldi (MPEG) for lending materials analyzed in this study; the Domingos de Jesus Rodrigues, curator Acervo Biológico da Amazônia Meridional–

ABAM of the Universidade Federal do Mato Grosso (UFMT/Sinop), the BIOTA Environmental Projects and Consulting LTDA for help with fieldwork logistics; Leandro Wronski, Jailson Xavier, Karll Cavalcante Pinto, and Renan Oliveira for their assistance in the field.

## Author Contributions

**Conceptualization:** Elciomar Araújo de Oliveira, Marcos Penhacek, José Gregório Martínez, Luís Reginaldo Ribeiro Rodrigues, Diego José Santana, Emil José Hernández-Ruz.

**Data curation:** Elciomar Araújo de Oliveira, Diego José Santana, Emil José Hernández-Ruz.

**Formal analysis:** Elciomar Araújo de Oliveira, Karen Larissa Auzier Guimarães, Emil José Hernández-Ruz.

**Funding acquisition:** Elciomar Araújo de Oliveira, Leandro Alves da Silva.

**Investigation:** Elciomar Araújo de Oliveira, Leandro Alves da Silva, Elvis Almeida Pereira Silva, Karen Larissa Auzier Guimarães, Marcos Penhacek, José Gregório Martínez, Luís Reginaldo Ribeiro Rodrigues, Diego José Santana, Emil José Hernández-Ruz.

**Methodology:** Elciomar Araújo de Oliveira, Elvis Almeida Pereira Silva, Karen Larissa Auzier Guimarães, Marcos Penhacek, José Gregório Martínez, Luís Reginaldo Ribeiro Rodrigues, Diego José Santana, Emil José Hernández-Ruz.

**Project administration:** Emil José Hernández-Ruz.

**Resources:** Luís Reginaldo Ribeiro Rodrigues, Diego José Santana, Emil José Hernández-Ruz.

**Software:** Elciomar Araújo de Oliveira, Elvis Almeida Pereira Silva, Diego José Santana.

**Supervision:** Luís Reginaldo Ribeiro Rodrigues, Diego José Santana, Emil José Hernández-Ruz.

**Validation:** Elciomar Araújo de Oliveira, Leandro Alves da Silva, Elvis Almeida Pereira Silva, Marcos Penhacek, José Gregório Martínez, Luís Reginaldo Ribeiro Rodrigues, Diego José Santana, Emil José Hernández-Ruz.

**Visualization:** Elciomar Araújo de Oliveira, Leandro Alves da Silva, Marcos Penhacek, José Gregório Martínez, Luís Reginaldo Ribeiro Rodrigues, Diego José Santana, Emil José Hernández-Ruz.

**Writing – original draft:** Elciomar Araújo de Oliveira.

**Writing – review & editing:** Elciomar Araújo de Oliveira, Leandro Alves da Silva, Elvis Almeida Pereira Silva, Karen Larissa Auzier Guimarães, Marcos Penhacek, José Gregório Martínez, Luís Reginaldo Ribeiro Rodrigues, Diego José Santana, Emil José Hernández-Ruz.

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
