## [Decision Letter · Decision Letter 0]

10 Oct 2019

PONE-D-19-16550

Four new species of Pristimantis Jiménez de la Espada, 1870 (Anura: Craugastoridae) in the eastern Amazon

PLOS ONE

Dear Dr. Elciomar Araújo de Oliveira,

Thank you for submitting your manuscript to PLOS ONE. After careful consideration, we feel that it has merit but does not fully meet PLOS ONE’s publication criteria as it currently stands. Therefore, we invite you to submit a revised version of the manuscript that addresses the points raised during the review process.

Dear Authors,

First of all I apologise for the long time necessary to have this first reply. This was due, partly, to the difficulties in find available reviewers. In particular, one reviewer accepted to review the manuscript but never finalized it.

Therefore, this manuscript  has been reviewed by one expert reviewer.

The good new is that he/she found the manuscript very interesting and worth of note. He/she also made a lots of comments and request of modification that I think may help you to improve the manuscript.

I agree with the reviewer comments and also I have few, minor suggestion listed below:

- L 338, "mitochondrial strains". Please change in "mitochondrial haplogroups" (here and elsewhere).

- Table 1 legend. "diagonal lower" please change in "below diagonal".

- Table 1 legend. "diagonal superior" please change in "above diagonal".

- L297 "We recovered three main divergences..." please reword

- L341 "16s" . Please change in "16S".

- Legend Fig. 3. "16S mtDNA (A)" please consider to delete mtDNA here.

We would appreciate receiving your revised manuscript by 9th january 2020. To enhance the reproducibility of your results, we recommend that if applicable you deposit your laboratory protocols in protocols.io, where a protocol can be assigned its own identifier (DOI) such that it can be cited independently in the future. For instructions see: http://journals.plos.org/plosone/s/submission-guidelines#loc-laboratory-protocols

We look forward to receiving your revised manuscript.

Kind regards,

Riccardo Castiglia

Academic Editor

PLOS ONE

**Journal Requirements:**

2. We note that  Figure(s) 1 in your submission contain [map/satellite] images which may be copyrighted. All PLOS content is published under the Creative Commons Attribution License (CC BY 4.0), which means that the manuscript, images, and Supporting Information files will be freely available online, and any third party is permitted to access, download, copy, distribute, and use these materials in any way, even commercially, with proper attribution. For these reasons, we cannot publish previously copyrighted maps or satellite images created using proprietary data, such as Google software (Google Maps, Street View, and Earth). For more information, see our copyright guidelines: http://journals.plos.org/plosone/s/licenses-and-copyright.

a) You may seek permission from the original copyright holder of Figure(s) [#] to publish the content specifically under the CC BY 4.0 license.  

**Comments to the Author**

1. Is the manuscript technically sound, and do the data support the conclusions?

Reviewer #1: Partly

2. Has the statistical analysis been performed appropriately and rigorously? 

Reviewer #1: Yes

3. Have the authors made all data underlying the findings in their manuscript fully available?

Reviewer #1: Yes

4. Is the manuscript presented in an intelligible fashion and written in standard English?

Reviewer #1: Yes

5. Review Comments to the Author

Reviewer #1: The article is very interesting, and I think the species are valid. However, some points of the article need to be developed and described better. The diagnosis, descriptions, and variation must be adjusted, and more detailed. You must follow the nomenclature used for the descriptions of this group, since in the article the language used is different from what has been traditionally proposed. On the other hand, you should better develop some topics in the discussion, and include other species available in the genbank. For all the above I have decided to make major changes.

6. PLOS authors have the option to publish the peer review history of their article (what does this mean?). If published, this will include your full peer review and any attached files.

Reviewer #1: No

---

## [Author Response · Author response to Decision Letter 0]

10 Feb 2020

Rebuttal letters referring to ms ID PONE-D-19-16550

Dear Dr. Riccardo Castiglia

Academic Editor of PLOS ONE

We are thankful for the decision on our study. Please find attached our paper “Four new species of Pristimantis Jiménez de la Espada, 1870 (Anura: Craugastoridae) in the eastern Amazon”, which we are resubmitting for consideration for publication in PLOS ONE. We appreciate the time of reviewers and editors and we believe suggestions have contributed tremendously to the quality of our manuscript. Below we specify, point-by-point, how we dealt with suggestions.

Please let us know if anything else is needed.

Best regards,

Elciomar Araújo de Oliveira

Below, you will find our responses for the academic editor suggestions:

- L 338, "mitochondrial strains". Please change in "mitochondrial haplogroups" (here and elsewhere). 

R. Modified.

- Table 1, legend. "diagonal lower" please change in "below diagonal". 

R. Modified.

- Table 1, legend. "diagonal superior" please change in "above diagonal". 

R. Modified.

- L 297, "We recovered three main divergences..." please reword.

R. We rewrote this sentence.

- L 341, "16s". Please change in "16S".

R. Modified.

- Legend Fig. 3. "16S mtDNA (A)" please consider to delete mtDNA here.

R. Modified

Reply on Comments by Reviewer

Abstract

- L 67, Pristimantis gaigei (conspicillatus group) is distributed from southeastern Costa Rica to eastern Panama.

R. Panama distribution added.

- L 68, Tobago? Taboga is in Panama.

R. Hedges et al. 2008 use that name: Isla Taboga, which is in Panama.

Introduction

- L 114, Padial et al and Canedo et al. show that the group extends to Bolivia and eastern Brazil. 

R. Added the new regions.

- L 114, The eastern most distribution is in the Atlantic forest with vinhai, paulodutrai... and in Bolivia with fenestratus, peruvianus among others…

R. Added new information.

Morphological analyses

- L 232, I think it is appropriate to follow the nomenclature used for the structures by Duellman and Lerh 2009.

R. Modified to what is proposed by Duellman and Lerh 2009; and Padial and De La Riva 2009.

- L 233, See Duellman and Lerh 2009

R. Added more information, same as in Padial and De La Riva 2009.

- L 233, If you are following the widely used nomenclature in terranae, why choose only one group of characters? and does not use the names given to the structures (see Duellman and Lerh, 2009).

R. Modifications were made to adapt to what was proposed in the literature cited in the text. The chosen character group is also mentioned in taxonomy works, such as: Padial and De La Riva 2009, Maciel et al. 2012...).

- L 236, There are other kinds of folds too

R. Modified to the proposal by Duellman and Lehr 2009.

- L 238, This one can also be replaced by tubercles

R. Modified to the proposal by Duellman and Lehr 2009.

- L 244, See Duellman and Lerh, 2009. It can also be the widest part of the head; it is not always from the rictus.

R. Modified to follow what has been used recently: Lerh and May 2017, Shepack et al. 2016, and Lerh and Moravec 2017.

- L 257, It is necessary that you can check the presence and development of the testicles, or convoluted oviducts (usually this is done with a small lateral cut). Otherwise it is difficult to determine if the individual is a sub-adult or a juvenile, or even a male or female.

R. This form of sex determination is the same proposal in other works, as: Oliveira et al. 2017, Lerh et al. 2017, Rivera-Correa et al. 2016, Fouquet et al. 2013).

- L 257, Adult? Or Juvenile? Check S3 File

R. Adult males were identified by viewing the secondary characters (visualization of secondary sexual characters in adult individuals, such as the presence or absence of vocal slits, vocal sac and nuptial callus), when these characters were not visualized, the individuals were considered juveniles.

- L 257, This may indicate that your individuals may be poorly sexed.

R. This form of sex determination is the same proposal in other works, as: Oliveira et al. 2017, Lerh et al. 2017, Rivera-Correa et al. 2016, Fouquet et al. 2013).

Bioacoustics analyses

- L 275, In addition, I will ask you to enter these recordings into a national or international database, so that they can be consulted in other studies, see Dena et al, 2019.

R. The vocalizations will be deposited in a database.

- L 282, Length of pulses? Length of internote? Number of pulses? Please add. Please add, as well as a description of the call by species.

R. The parameters used follow the work of describing new species of Pristimantis, as: Kok et al. 2018, Shepack et al. 2016, Rivera-Correa et al. 2016, Fouquet et al. 2013, Maciel et al. 2012, Padial e De La Riva 2009.

Results

- L 299, In some places detect Portuguese

R. Checked.

Phylogenetic relationships and genetic distances

- L 311, Correct in the image "achantinus: achatinus"

R. Revised

- L 311, Why don't you include other terminals of conspicillatus group available at GenBank? Terraebolivaris, samaipatae, koehleri, skydmainos... see Canedo et al.2012. 

R. According to GenBank, there are no sequences of P. terraboliviaris, P. koehleri, P. samaipatae and P. skydmainos for COI, so we don't use sequences from the same species that we found for the 16S gene.

- L 311, More sequences of other species of the group are very important to see the relationships with the species that are available in GenBank. On the other hand this allows a better comparison.

R. There are many Pristimantis conspicillatus group that does not have sequences for the COI, so I didn't put it in the analysis.

- L 349, This needs detailed description.

R. New information was added in the legend.

- L 353, A table is not enough to describe the calls of these species, please include an item in each species description and detail briefly.

R. New information was added to the diagnosis topic for each species.

Systematics 

Allocation to species group

- L 378, You could improve the presentation of the figures: for example, center more the individuals, at least so that it looks completely and do not cut the heels, the snout, and the fingers.

R. Modified

- L 382, Make a single plate with the photographs of the holotypes.

R. Modified

- L 382, Make one plate with all holotypes, and add a lateral view of head. In addition, could you put the foot where the ventral part is best seen? as in pluvian and moa

R. Modified.

- L 387, 1 plate.

R. Modified

Diagnosis

- L 400, These comments are for all species: I think that the descriptions part for all species needs a lot more work. The Pristimantis are a very large group, and if a description does not have the level of detail, it will need re-descriptions in the future, or a better review of this group. 

R. We believe that in the diagnosis it is not necessary to add characteristics that are not diagnostic, because the genus is very conserved, so we put only those that can really differentiate one species from the other. However, we added information in the description of each new species. 

- L 400, Follow terminology of Duellman and Lerh 2009.

R. We follow the proposal by Duellman and Lehr (2009) and other authors, such as: Elmer and Cannatella (2008); Kok and Kalamandeen (2008); and Padial and De la Riva (2009). 

- L 400, Remember to write in passive voice, in diagnosis, description of holotype...

R. We have followed the same description previous published on the Plos One.

- L 406, The males? With vocal slits, bocal sac? Nuptial pad?

R. These are not diagnosis in relation to the other species already described, mainly in this work, since all of them vocal sac and callus present.

- L 406, Vomer? Form and presence?

R. They are not diagnostic features, since they all have the same pattern.

- L 409, What happen with heel, and eyelid, scapular, and other folds? 

R. They are not diagnostic features.

- L 409, The subarticular tubercles are projected? Low?

R. Projected. Modified in the text

- L 411, Could be more detailed

R. New details have been inserted.

- L 412, I do not think it relevant to be part of the diagnosis, it will not be the only species distributed in this place.

R. Removed.

Comparison with other species

- L 423, Why was it not included in phylogeny? there are sequences available in GenBank...

R. There are no sequences of P. terraboliviaris, P. koehleri, P. samaipatae and P. skydmainos for COI in the GenBank.

- L 431, Scapular? Sacral? Middorsal?

R. The tubercles are spread throughout the dorsal region (scapular, middle of the body, lateral and supraocular). Information added in the ms.

Description of the holotype

- L 434, Follow terminology of Duellman and Lerh 2009.

R. In addition to following Duellman and Lerh 2009, we follow the recent articles describing new species of Pristimantis (Navarrete et al. 2016, Lehr and Moravec 2017, and Sánchez-Nivicela et al. 2018)

- L 434, A much more detailed description is necessary. In Pristimantis this level of detail is very necessary, given the number of species and the taxonomic problems that already exist. This comment applies to other new species.

R. New details have been inserted.

- L 436, Folds?

R. Dorsolateral fold absent.

- L 449, How many?

R. One.

- L 449, Are there tubercle on the heel?

R. Absent.

- L 454, What happen with outer tarsal tubercles?

R. Outer tarsal tubercles absent

- L 455, You never mention what happens with vocal slits, nuptial pads, vomers? Number of teeth on dentigerous process, heel tubercles, eyelid tubercle, outer tarsal tubercles... 

R. We follow the model of the table describing new species of Pristimantis in Padial and De La Riva 2009 and Maciel et al. 2012, where we use the same information available.

- L 455, What is the reason for not following the standard terminology in the group see Duellman and Lerh.

R. We follow what has been proposed in the latest descriptions of Pristimantis (Navarrete et al. 2016, Lehr and Moravec 2017, and Sánchez-Nivicela et al. 2018), Additionally Duellman and Lerh (2009).

- L 457, See Duellman and Lerh 2009, different kinds of folds

R. Modified. 

- L 458, Sometimes are represented by a row of tubercles in the same position.

R. Modified.

- L 463, Enter the data to a library database: Colección de sonidos ambientales IavHv Colombia and Foneteca Neotropical GACQUESVIELLIARD in São Paulo (these are international). It is very important that these recordings be public for future comparisons. Review Dena et al, 2019. How much are we losing in not depositing anuran songs recordings in scientific collection? Journal Bioacustic. DOI: 10.1080/09524622.2019.1633567

R. Recordings deposited as request

- L 463, It is very important that they describe the call, since a table will not describe it completely. For example, the first note in some calls has lower parameters. number of harmonics, see Kohler et al. 2017. The use of bioacoustics in anuran taxonomy...

R. I added new information in the diagnosis of each species.

- L 463, Number of individuals? I don’t understand...

R. Revised.

- L 463, Please report the pulse length and interpulse length, internote

R. In recent works and those we followed, these data are not presented, so we do not use: Kok et al. 2018, Shepack et al. 2016, Rivera-Correa et al. 2016, Fouquet et al. 2013, Maciel et al. 2012, and Padial and De La Riva 2009.

- L 463, You analyze different populations?

R. As reported in the vocalization table, for some species, yes.

- L 473, What happen with chin, hidden parts of the thigh, groin?

R. We added new information at the end of the paragraph.

- L 474, In dorsal and ventral view?

R. Yes. Modified.

- L 474, Canthal stripe? Supratympanic stripe? Other facial markings? See Duellman and Lerh 2009.

R. Modified.

- L 475, The name is canthal stripe.

R. Modified.

- L 475, You don’t follow appropriated terminology? See Duellman and Lerh 2009

R. Modified.

- L 489, This is not representative for the variation, what happen between males and females? are there differences? in which individuals? between the same sex there are variation in folds, tubercles? shapes? Vomer? Folds? Color? Groin? Texture? ....

R. We add the difference between males and females, as well as other information.

- L 489, A table with the averages of the measures for males and females

R. We add the average between males and females in the body of the text.

- L 491, There are many kinds of dorsal tubercles.

R. Present tubercle scattered throughout the dorsal region (scapular lateral middle of the body).

- L 516, The same comments for the diagnosis, description, and variation. These items need a lot of work and a better, more detailed description. You must follow the nomenclature used in Pristimantis.

R. We added new information on the topics of diagnoses, description of the holotype, variation, ....

- L 523, Make one plate with all holotypes, and add a lateral view of head. In addition, could you put the foot where the ventral part is best seen? as in pluvian and moa

R. Modified.

Diagnosis

- L 543, Describe better see comments in giorgii

R. We added new information on the topics of diagnoses, description of the holotype, variation, ....

Comparison with other species

- L 578, See other comments

R. We have inserted new information on this topic in all species.

Description of the holotype

- L 583, Describe better see comments in giorgii

R. We have inserted new information on this topic in all species.

Color in Life

- L 613, Describe better see comments in giorgii

R. We have inserted new information on this topic in all species.

Variation

- L 631, Describe better see comments in giorgii

R. We add the difference between males and females, as well as other information.

- L 662, The same comments for the diagnosis, description, and variation. These items need a lot of work and a better, more detailed description. You must follow the nomenclature used in Pristimantis.

R. We added new information on the topics of diagnoses, description of the holotype, variation, ....

- L 664, Make one plate, and lateral view of head

R. Modified, but we don’t put an image of the side view of the head, following what was published in other works: Navarrete et al. 2016; Shepack et al. 2016; Brito et al. 2017; and Lehr and Moravec 2017.

Diagnosis

- L 687, Describe better see comments in giorgii

R. We added new information on the topics of diagnoses, description of the holotype, variation, ....

Description of the holotype

- L 726, Describe better see comments in giorgii

R. We added new information on the topics of diagnoses, description of the holotype, variation, ....

Color in Life

- L 756, Describe better see comments in giorgii

R. We added new information on the topics of diagnoses, description of the holotype, variation, ....

Variation

- L 774, Describe better see comments in giorgii

R. We added new information on the topics of diagnoses, description of the holotype, variation, ....

Distribution, ecology and habitat

- L 803, The same comments for the diagnosis, description, and variation. These items need a lot of work and a better, more detailed description. You must follow the nomenclature used in Pristimantis.

R. New information was inserted in all the mentioned topics. We follow the nomenclature of Duellman and Lehr 2009, in addition to other more recent authors: Navarrete et al. 2016, Lehr and Moravec 2017, and Sánchez-Nivicela et al. 2018.

- L 805, Make one plate, and lateral view of head

R. Modified, but we don’t put an image of the side view of the head, following what was published in other works: Navarrete et al. 2016; Shepack et al. 2016; Brito et al. 2017; and Lehr and Moravec 2017.

Diagnosis

- L 823, Describe better see comments in giorgii

R. We have inserted new information on this topic in all species.

Description of the holotype

- L 871, Describe better see comments in giorgii

R. We have inserted new information on this topic in all species.

Variation

- L 909, Describe better see comments in giorgii

R. We have inserted new information on this topic in all species.

Discussion

- L 931, What about conspicillatus diversity? Apparently, a lot of research is needed on new species, cryptic diversity, speciation processes, biogeography...

PLEASE EXPAND, YOU HAVE WRITTEN ALL THE PAPER OF THIS AND IN THE DISCUSSION, YOU ONLY HAVE A SMALL SPACE FOR THIS.

R. New information was inserted to agree the reviewer's suggestion.

- L 938, For vertebrates?

R. Terrestrial species (animal, fungi, plants, Protozoa …)

- L 941, Repeat in introduction

R. This phrase was partially repeated in the introduction, but at this point in the discussion we say that most species were described for the Andes, and in the other sentence, we have already addressed the heterogeneity of the environment in the role of diversification of the species of Pristimantis. So, we believe that you don't need to remove that part.

- L 957, This can also be the product of our inability to detect morphological differences, we need to explore other sources of characters that are not external morphology. For example, in many cases the solution has been morphometry.

R. We agree with the reviewer's comment and modifications were made.

- L 964, There are other explanations for this, I think you are oversizing this finding. What happens if you put more species? and more genes?

R. Modified to “Phylogenetic reconstruction shows that P. pluvian, P. fenestratus, P. sp. (municipality of Juruti, PA) and P. sp. (municipality of Borba, AM) belong to a hard polytomy, which may be common in organisms experiencing rapid speciation but with incomplete lineage sorting in mtDNA”.

- L 969, I think you can broaden this topic a bit, about the importance of knowing biodiversity at this time of global declines.

R. New information has been inserted.

---

## [Editor Report · Decision Letter 1]

20 Feb 2020

Four new species of Pristimantis Jiménez de la Espada, 1870 (Anura: Craugastoridae) in the eastern Amazon

PONE-D-19-16550R1

Dear Dr. Elciomar Araújo de Oliveira,

We are pleased to inform you that your manuscript has been judged scientifically suitable for publication and will be formally accepted for publication once it complies with all outstanding technical requirements.

With kind regards,

Riccardo Castiglia

Academic Editor

PLOS ONE

Additional Editor Comments (optional):

Thank you for your revised version, and compliments for your work.

best wishes 

Riccardo Castiglia
---

## [Editor Report · Acceptance letter]

25 Feb 2020

PONE-D-19-16550R1 

Four new species of *Pristimantis* Jiménez de la Espada, 1870 (Anura: Craugastoridae) in the eastern Amazon 

Dear Dr. de Oliveira:

I am pleased to inform you that your manuscript has been deemed suitable for publication in PLOS ONE. Congratulations! Your manuscript is now with our production department. 

With kind regards,

on behalf of

Dr. Riccardo Castiglia 

Academic Editor

PLOS ONE